# A conformational switch in clathrin light chain regulates lattice structure and endocytosis at the plasma membrane of mammalian cells

Kazuki Obashi [1], Kem A. Sochacki[1], Marie-Paule Strub[1] & Justin W. Taraska [1] ✉

Conformational changes in endocytic proteins are regulators of clathrin-mediated endocytosis. Three clathrin heavy chains associated with clathrin light chains (CLC) assemble into triskelia that link into a geometric lattice that curves to drive endocytosis. Structural changes in CLC have been shown to regulate triskelia assembly in solution, yet the nature of these changes, and their effects on lattice growth, curvature, and endocytosis in cells are unknown. Here, we develop a new correlative fluorescence resonance energy transfer (FRET) and platinum replica electron microscopy method, named FRET-CLEM. With FRET-CLEM, we measure conformational changes in clathrin at thousands of individual morphologically distinct clathrin-coated structures. We discover that the N-terminus of CLC repositions away from the plasma membrane and triskelia vertex as coats curve. Preventing this conformational switch with chemical tools increases lattice sizes and inhibits endocytosis. Thus, a specific conformational switch in the light chain regulates lattice curvature and endocytosis in mammalian cells.

Clathrin-mediated endocytosis is the primary internalization pathway in eukaryotic cells and is key to many processes, including nutrient uptake, excitability, signaling, and the recycling of membrane components[1]. Over 50 proteins have been implicated in this process[2,3]. Determining the nanoscale localizations[4,5], numbers[6,7], accumulation dynamics[8,9], interactions[1], and conformations[10,11] of these proteins in cells is important for understanding how the endocytic machinery operates in health and disease.

Assembly and curvature of clathrin lattices are required steps to build cargo-loaded vesicles[12–14]. The basic unit of the clathrin lattice is the clathrin triskelion[15,16]. This six protein complex is composed of three clathrin heavy chains and three smaller clathrin light chains (CLCs) that form a three-legged pinwheel (Supplementary Fig. 1a)[17]. In non-neuronal cells, the stoichiometry of heavy chains to light chains might not always be one-to-one[18]. The heavy chains provide the backbone, and light chains are thought to regulate the assembly of the

lattice and effect its mechanical properties[16]. Work both in vitro and in cells has indicated its important role[19–23], yet the structural, and functional actions of the light chains are not fully understood[16,24].

In vitro, clathrin triskelia assemble into spherical cages[13]. Clathrin light chain inhibits this assembly[25,26]. Light chain binds to heavy chain through multiple interactions[16]. One specific interaction occurs through an acidic patch (EED) near the light chain's N-terminus and the heavy chain's knee (Supplementary Fig. 1a)[27]. This interaction prevents cage assembly by regulating the heavy chain knee conformation (Supplementary Fig. 1b)[17]. If CLC binds to the heavy chain through these N-terminal contacts, CLC adopts an extended conformation, stabilizing a straight conformation of the heavy chain knee. The straight conformation cannot assume the angles needed for cage assembly. However, if this interaction is prevented, the heavy chain knee is free to move, allowing the knee to bend and assemble as a cage. Thus, conformational switching of CLC is

[1]Biochemistry and Biophysics Center, National Heart, Lung, and Blood Institute, National Institutes of Health, 50 South Drive, Building 50, Bethesda, MD 20892, USA. ✉e-mail: justin.taraska@nih.gov

proposed to control the assembly of lattices[17]. Yet, the nature of these structural changes, and their effects on lattice growth, and curvature at the membrane of living cells are mostly unknown.

Recent super-resolution imaging studies have mapped changes in protein locations at distinct stages of endocytosis at the scale of tens of nanometers[4,5]. Proteins are, however, regulated by sub-ten nanometer intra- and intermolecular conformational changes and binding events. To understand endocytosis, these conformational changes need to be determined[3,13]. This is a major gap in understanding endocytosis. Yet, the precision of super-resolution imaging cannot measure structural changes at these scales. Fluorescence (or Förster) resonance energy transfer (FRET), however, can fill this gap, mapping molecular interactions and conformational changes within and between proteins[28]. FRET efficiency depends on the spectral overlap between the donor and acceptor, distances between the dipoles, and their relative orientations to each other. Specifically, FRET occurs when a donor fluorophore and acceptor are generally separated by less than 10 nm[29]. While FRET measurements are usually made with diffraction-limited imaging, super-resolved FRET methods have been reported[30]. The resolution, labeling density, and colors possible in these experiments, however, are not able to discriminate morphological differences in small organelles such as clathrin-coated structures (CCSs). For example, past FRET analysis of the protein organization of CCSs in yeast was limited to a late stage of endocytosis trapped by drug treatments[31]. Thus, existing methods cannot readily relate changes in protein conformation generated from FRET to the morphological stages of clathrin-coated pits as they grow and curve.

To overcome this gap, we turned to correlative light and electron microscopy (CLEM)[32]. CLEM can directly map fluorescence signals to single nanoscale cellular structures visualized in paired electron microscopy (EM) images[33]. Platinum replica transmission EM (PREM) of unroofed cell plasma membranes provides a uniquely high contrast, high-resolution, and wide-field view of the inner plasma membrane of cells[34]. PREM has been combined with diffraction limited[35,36], super-resolution[4,37–40], and polarized total internal reflection fluorescence (TIRF)[41] microscopy to investigate molecular mechanisms of endocytosis, exocytosis, and the cortical cytoskeleton.

Here, we develop a correlative lifetime based-FRET (FLIM-FRET) and PREM method, named FRET-CLEM. This method allows us to measure changes in protein interactions and conformations at distances less than 10 nm at single structurally-defined organelles in single cells. We investigate the conformational changes in clathrin light chain by mapping distance changes both parallel and perpendicular to the plasma membrane. We find that the N-terminus of CLC moves away from both the CLC C-terminus and the plane of the plasma membrane as clathrin sites gain curvature. To determine the mechanistic impact of these structural changes, we develop a method to directly manipulate the N-terminal position of the clathrin light chain using a chemically-inducible dimerization system. These manipulations were confirmed with FRET. With acute chemical perturbation, we show that inhibiting conformational changes in CLC's N-terminus increases clathrin lattice size, impairs maturation of clathrin structures at the plasma membrane, and inhibits transferrin endocytosis. Together, these data reveal a new conformational switch in clathrin light chain that regulates endocytosis in living cells.

## Results

### FRET-CLEM provides nanometer-scale spatial information at single clathrin sites

To determine sub-ten nanometer conformational movements during endocytosis, we established a new correlative FLIM-FRET and PREM method, which we named FRET-CLEM (Fig. 1). We chose monomeric EGFP and a dark yellow fluorescent protein, ShadowY[42], as an optimized FRET pair by comparing six potential fluorescent protein (FP) pairs (Supplementary Fig. 2a). Calculations from the emission and absorbance spectra of this pair resulted in an estimated $R_0$ value (distance of 50% FRET efficiency) of 60 Å (Supplementary Fig. 2b). In line with these calculations, tandemly-connected EGFP-ShadowY probes fused to CLC showed >50% FRET efficiency on the plasma membrane (Supplementary Fig. 3a–d). We used this construct as a positive FRET control. Next, to test whether FRET could be localized to single CCS, we performed FRET-CLEM measurements on HeLa cells expressing EGFP-CLC or EGFP-ShadowY-CLC (Fig. 1a). First, cells were unroofed to expose the inner surface of the plasma membrane and rapidly fixed[34]. These membranes provide uniquely high contrast, low background, ultra-thin samples for fluorescence imaging[32]. Unroofed cells were then imaged with FLIM and subsequently prepared for platinum replica electron microscopy (PREM). Past work has shown that these sample preparation steps do not measurably change the morphology and structure of the membrane and its associated endocytic organelles[4,37,43]. This correlative method allows us to assign single diffraction-limited fluorescent spots from a FLIM image to a single clathrin structure visualized in PREM (Fig. 1a). Because PREM has a high spatial resolution, single fluorescent spots in the FLIM image can be further classified according to the nanoscale structural features of the clathrin lattice (Fig. 1b)[43]. We then tested whether fluorescence lifetimes can be analyzed at single CCS resolution. With the total photon number detected by our FLIM acquisition parameters (Supplementary Fig. 3e–g, see also Methods), fluorescence lifetime decay curves determined from single CCSs were clearly separated between negative EGFP-CLC and positive EGFP-ShadowY-CLC FRET controls (Fig. 1c). Specifically, they could be fit with a bi-exponential where the FRET efficiency was determined for each CCS (Supplementary Fig. 3h)[44]. However, curve-fitting was suboptimal for small CCSs due to the limited number of photons[45]. Thus, to measure as many clathrin structures as possible across a cell, the mean fluorescence lifetime was used to estimate FRET efficiencies across all structures (Fig. 1d). These results show that FRET-CLEM can be used to generate FRET-based atomic-scale distances at single CCS resolution at morphologically distinct stages of endocytosis at the plasma membrane of mammalian cells.

### CLC N-terminal region moves away from the clathrin triskelion vertex

Next, we applied FRET-CLEM to study conformational changes in CLC at the plasma membrane. In vitro, clathrin triskelia assemble into empty cages and CLC regulates this assembly[27]. CLC has been proposed to take two different conformations when bound to triskelia, an extended and bent conformation (Fig. 2a)[17]. Here, we assume three models of CLC conformations in cells based on in vitro models (Fig. 2a). First, we assume that unassembled triskelion in the cytoplasm contains extended CLC and the spherical lattices in cells resembles in vitro assembled clathrin cages and contains bent CLC. Next, we propose three possible models for light chain dynamics during curvature. In the first, CLC does not change conformations after assembly on the plasma membrane. Here, CLC assumes the bent conformation regardless of the curvature stage of the lattice (flat, domed, spherical). In the second, the proportions of extended and bent CLC gradually shift as curvature increases during endocytosis. In the third, as yet undescribed conformations of CLC are present in flat and domed clathrin that switches to the bent conformation in spheres.

To test these models, we used FRET-CLEM. Specifically, the distances between the N-terminus and C-terminus of surrounding CLCs differ substantially between the extended and bent conformations according to structural models[17,46] (Fig. 2b, c, and Supplementary Fig. 1). Although the expression levels of the transfected probes, the ratio between endogenous and transfected CLCs, and the fraction of heavy chains binding to CLC could affect FRET efficiency, FRET

efficiencies of the extended conformation are always larger than that of bent conformation when the degree of these factors are stable (Supplementary Fig. 4). This condition is satisfied when comparisons are made within a single cell. Thus, if the population of extended and bent conformations change on the plasma membrane during lattice assembly, FRET between the N- and C-terminus of CLCs is predicted to likewise change (model 2 in Fig. 2a). Alternatively, if the conformations do not change, FRET should remain the same across all clathrin subtypes at the plasma membrane (model 1 in Fig. 2a).

To compare models 1 and 2, we performed FRET-CLEM measurements on HeLa cells expressing EGFP-CLC, or EGFP-CLC and CLC-ShadowY (Fig. 2d and Supplementary Fig. 5a, b). FPs are attached to CLC through a flexible linker (Supplementary Fig. 6, see also Methods).

The FP-attached CLCs were the dominant species within cells with our transfection condition (Supplementary Fig. 7, see also Methods). Mean fluorescence lifetimes from single CCSs were analyzed by grouping them according to clathrin curvature classes determined from PREM images (flat, domed, and sphere; Fig. 1b). There was no clear relationship between mean fluorescence lifetimes and photon counts (Supplementary Fig. 8). In cells expressing both EGFP-CLC and CLC-ShadowY, fluorescence lifetimes increased as clathrin lattices curved (Fig. 2d). This substantial increase did not occur when expressing only EGFP-CLC. The neuronal isoform of CLC which has an insertion of residues near the C-terminus[16] and was used in the previous in vitro study[17] showed similar lifetime changes (Supplementary Fig. 9). Furthermore, similar results were obtained in SK-MEL-2 human cells

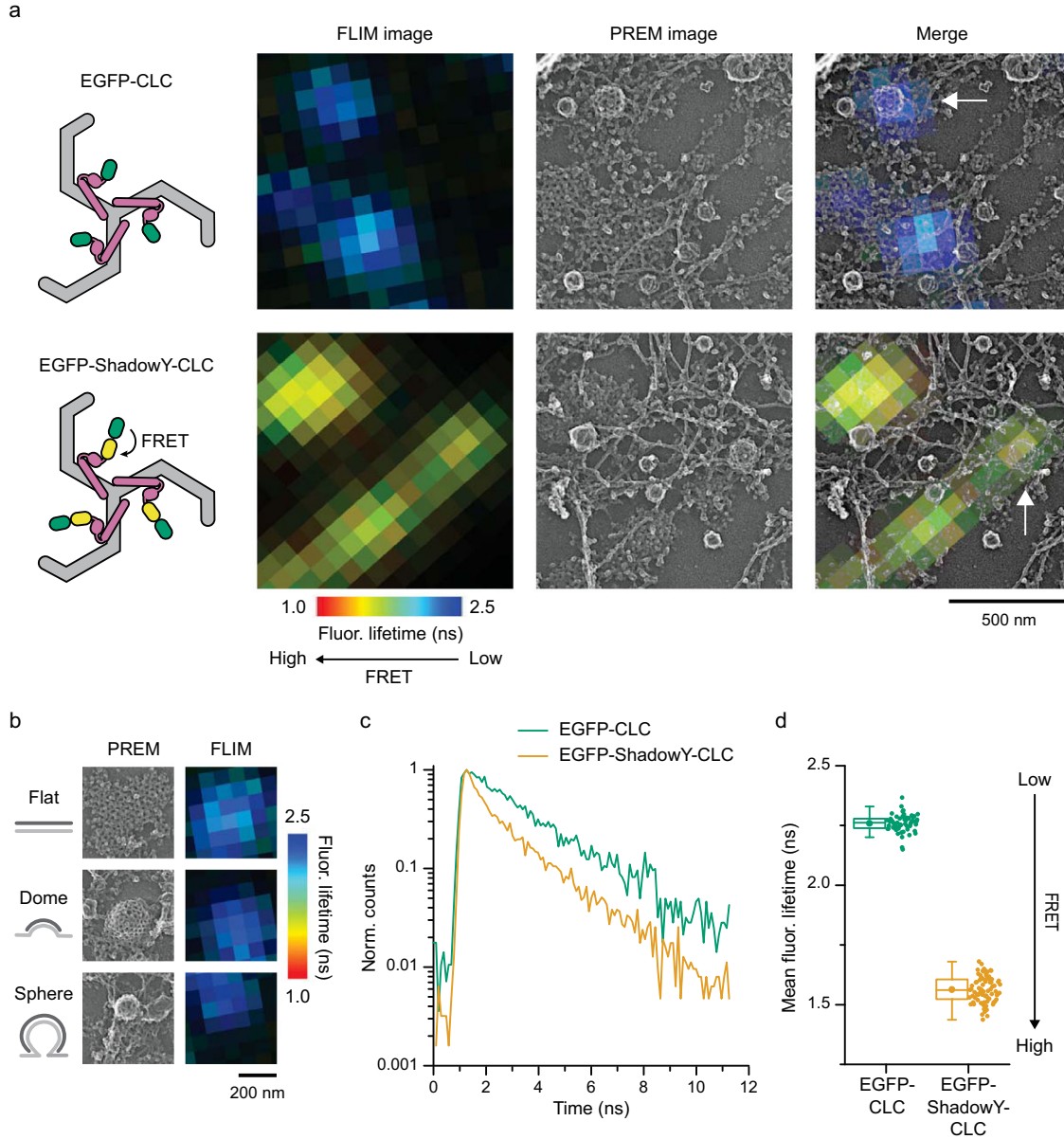

**Fig. 1 | FRET-CLEM. a** Correlative FLIM-FRET and PREM images of unroofed membranes of HeLa cells expressing EGFP-CLC (top) or EGFP-ShadowY-CLC (bottom). FLIM images (left; photon counts are represented by brightness and fluorescence lifetimes are represented by pseudo color), PREM images (center), and merge images (right). *n* = 1 cell for each condition. Scale 500 nm. **b** PREM images of CCSs on an unroofed membrane of a HeLa cell expressing EGFP-CLC that were classified as flat, domed, or sphere and corresponding areas from a FLIM image.

*n* = 1 cell. Scale 200 nm. **c** Fluorescence lifetime decays from single CCSs indicated by arrows in panel a. **d** Mean fluorescence lifetimes from single CCSs on an unroofed membrane of HeLa cells expressing EGFP-CLC or EGFP-ShadowY-CLC. *n* = 58 CCSs from 1 cell (EGFP-CLC) and 81 CCSs from 1 cell (EGFP-ShadowY-CLC). For box plots, box is interquartile range, center line is median, center circle is mean, whiskers are minimum and maximum data points with a coefficient value of 1.5. Source data are provided as a Source Data file.

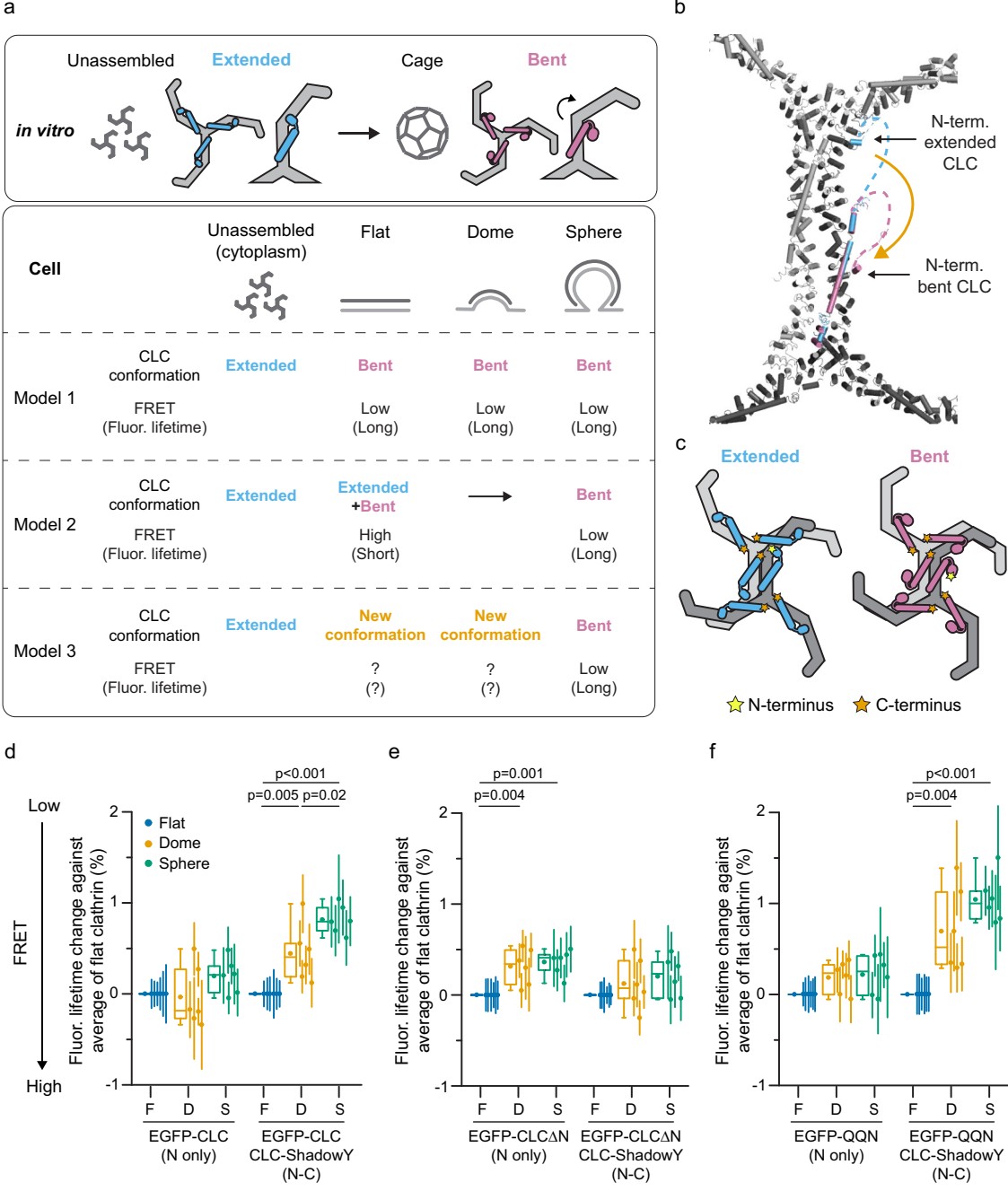

**Fig. 2 | CLC conformational changes in cells. a** Schematic models of CLC conformation at different assembly states in vitro or in living cells and their expected FRET efficiencies between the N- and C-terminus of CLCs. **b** A structural model with the assumption that either extended (light blue) or bent conformations (magenta) of CLCs assemble into the lattice. The model is based on PDB 3LVG and 6WCJ. 3LVG is overlaid with 6WCJ. **c** Schematic models of two assembled triskelia with extended (left) or bent CLCs (right). A CLC N-terminus position (yellow) and C-terminus positions of surrounding CLCs (orange) are shown. **d** FRET-CLEM was performed on HeLa cells expressing either EGFP-CLC, or EGFP-CLC and CLC-ShadowY. Mean fluorescence lifetimes from single CCSs were analyzed by categorizing them according to lattice structures (flat, domed and sphere) and they were compared to the average values of flat structures. $n = 6$ cells from 3 experiments (EGFP-CLC) and $n = 6$ cells from 4 experiments (EGFP-CLC and CLC-ShadowY). **e** FRET-CLEM on HeLa cells expressing either EGFP-CLCΔN, or EGFP-CLCΔN and CLC-ShadowY. $n = 6$ cells from 5 experiments (EGFP-CLCΔN) and $n = 6$ cells from 4 experiments (EGFP-CLCΔN and CLC-ShadowY). **f** FRET-CLEM on HeLa cells expressing either EGFP-QQN (QQN mutant of CLC), or EGFP-QQN and CLC-ShadowY. $n = 6$ cells from 3 experiments for each condition. One-way ANOVA, then Tukey's test. Each dot is from one cell experiment and errors are SE. For box plots, box is interquartile range, center line is median, center circle is mean, whiskers are minimum and maximum data points with a coefficient value of 1.5. Source data are provided as a Source Data file.

indicating that these changes were not cell-type specific (Supplementary Fig. 10a–c). In living cells, FRET efficiencies between EGFP-CLC and CLC-ShadowY were higher on CCSs than in the cytosol (Supplementary Fig. 11). However, unlike previous measurements in solution[17], our FRET measurements reflected both intra- and inter-triskelia FRET. Next, we measured EGFP-CLCΔN which is a truncation mutant of the flexible N-terminal domain (residues 1–89) (Fig. 2e and Supplementary Fig. 5c, d). In this mutant, EGFP would be bound near the heavy chain binding helix[15]. For this truncation, fluorescence lifetimes did not substantially change across different lattice states. These data indicate that the N-terminal position of CLC undergoes a specific conformational switch at clathrin lattices on the plasma membrane during

endocytosis. These data are not consistent with model 1. Furthermore, fluorescence lifetimes from cells expressing EGFP-CLC and ShadowY-CLC changed across different stages (Supplementary Fig. 5g, h). This supports the movement of the CLC N-terminal region. Next, to further test model 2, we measured QQN mutants of clathrin light chain (residues 20–22 were substituted from EED to QQN) (Fig. 2f and Supplementary Fig. 5e, f). QQN mutants are deficient in binding to clathrin heavy chain near the triskelion vertex due to a loss of a negatively-charged patch at the N-terminus of the light chain (Supplementary Fig. 1a)[27]. Thus, they are inhibited from adopting the proposed extended conformation[17]. Here, if model 2 is correct, FRET across the structural states should differ between wild-type and QQN mutants. Specifically, QQN mutants are expected to show a smaller displacement in conformations and smaller changes in fluorescence lifetime as the lattice curves. However, in these mutants, we found similar fluorescence lifetime changes as in the wild-type protein (Fig. 2d,f). This similarity in fluorescence lifetime change was observed among the range of expression levels obtained in our experimental conditions (Supplementary Fig. 12). These data do not support model 2. Thus, we conclude that CLC structural movements at the plasma membrane cannot be described by current in vitro models (model 1 or 2).

### CLC N-terminal region moves away from the plasma membrane

Next, we measured FRET between EGFP positioned in different domains of clathrin light chain and the membrane resident dark FRET-based quencher dipicrylamine (DPA) (Supplementary Fig. 2c)[47]. Because the plane of the membrane is fixed, FRET between EGFP and DPA provides relative distances perpendicular to the plane of the plasma membrane[48,49]. Specifically, when EGFP sits near the membrane, the FRET efficiency is high and fluorescence lifetimes are short (Fig. 3a). With DPA, CLC-EGFP showed the shortest lifetime (Fig. 3b,c). This indicates that the C-terminus of CLC is located close to the plasma membrane. In contrast, EGFP-CLC showed the longest fluorescence lifetime. EGFP-CLCΔN showed an intermediate lifetime. From these data, we can position segments of CLC relative to the plane of the plasma membrane (Fig. 3d). Unlike classic models[17], we find that the N-terminus of CLC bound to clathrin-coated structures at the plasma membrane is located farther into the cytoplasm than the proximal leg of clathrin heavy chain.

To investigate conformational changes during lattice curvature, we performed FRET-CLEM measurements with DPA and compared those measurements to morphological changes in clathrin lattice (Fig. 3e and Supplementary Fig. 13). In these experiments, fluorescence lifetimes of EGFP-CLC increased as lattices curved, and the degree of change was larger than those for CLC-EGFP. Similar results were obtained for cells expressing SNAP-CLC-EGFP or EGFP-CLC-SNAP (Supplementary Fig. 14). In addition, in SK-MEL-2 cells, fluorescence lifetimes of EGFP-CLC increased as lattices curved, and the degree of change was similar to that of HeLa cells (Supplementary Fig. 10d, e). These data are consistent with a model where the N-terminal position of CLC changes during curvature. Thus, the combined results from EGFP-ShadowY and EGFP-DPA FRET indicate that the N-terminus of CLC extends away from the CLC C-terminus (triskelion vertex) and the plane of the plasma membrane during endocytosis (Fig. 3f).

### Conformational changes in CLC regulate lattice structure and endocytosis

Next, we tested whether these conformational changes in CLC regulate clathrin-mediated endocytosis. To manipulate the CLC N-terminal position directly, we used a rapamycin-inducible FKBP/FRB dimerization system[50]. Specifically, we employed the T2098L mutant of FRB which is heterodimerized to FKBP by the rapamycin analog, AP21967[51]. We attached FKBP to the N-terminus of CLC, and one or two FRBs (FRB×2) to the C-terminus of the membrane-bound PH domain from

PLCδ1 (PH)[52] (Fig. 4a). We made both FRB and FRB×2 probes to extend the reach of the system. To confirm that the N-terminal position of FKBP-EGFP-CLC changes relative to the plasma membrane after dimerization, we measured EGFP fluorescence lifetimes without or with AP21967 in the presence of DPA (Fig. 4b,c). Changes in the position of EGFP can be estimated by comparing the difference in fluorescence lifetimes between the control and DPA-treated membranes (Fig. 4d). For both probes, differences in EGFP lifetimes with AP21967 treatment were larger than those without AP21967. These results indicate that the CLC N-terminus moves towards the plasma membrane as a result of FKBP/FRB dimerization. Because PH-miRFP-FRB×2 showed larger lifetime change (Fig. 4d) and stronger accumulation with AP21967 than PH-miRFP-FRB (Supplementary Fig. 15a), we used PH-miRFP-FRB×2 (or PH-mCherry-FRB×2) to alter the structure of clathrin light chain in living cells.

First, using our FKBP/FRB constructs, we investigated the impact of moving the CLC N-terminus towards the plasma membrane on the structure of clathrin lattices[43]. Here, unroofed membranes from cells expressing FKBP-EGFP-CLC and PH-miRFP-FRB×2 and treated without or with AP21967 were imaged with PREM (Supplementary Fig. 16). Unlike the previous homodimerization system[53], clathrin lattices were not noticeably distorted. The average size of all visible flat and domed clathrin structures increased with AP21967 treatment [$31463 \pm 859\,nm^2$ (ctrl) and $42919 \pm 2278\,nm^2$ (AP) for flat, $29821 \pm 576\,nm^2$ (ctrl) and $36471 \pm 2862\,nm^2$ (AP) for domed, $15311 \pm 588\,nm^2$ (ctrl) and $17303 \pm 691\,nm^2$ (AP) for sphere] (Fig. 5a-c). We conclude that manipulation of the CLC N-terminal position directly effects the structure of clathrin lattices.

Next, to investigate how changing CLC conformations effects CCS assembly and maturation, we performed live cell time lapse evanescent field imaging. Cells expressing FKBP-EGFP-CLC and PH-miRFP-FRB×2 were imaged with total internal reflection fluorescence microscopy (TIRF) in the absence or presence of AP21967 (Fig. 5d, e). To quantitate CCS dynamics, we analyzed the residence times of FKBP-EGFP-CLC spots on the plasma membrane. The residence times became shorter after AP21967 addition. This indicates that tethering the CLC N-terminal position towards the plasma membrane likely facilitates the disassembly of CCSs.

Finally, we investigated whether manipulation of the CLC N-terminal position effects transferrin endocytosis in whole cells (Fig. 5f, g and Supplementary Fig. 17a, b). As a control for binding of FRB to the CLC N-terminus, a cytosolic probe, mCherry-FRB×2, was used. Furthermore, as a control of clathrin tethering to the plasma membrane, and also FRB accumulation on CCSs, a C-terminal-linked FKBP probe, CLC-FKBP-EGFP, was tested. For all three probe pairs, FKBP/FRB dimerization was confirmed by clustering of FRB at CCSs (Supplementary Fig. 15b) and changes in FRET efficiency between EGFP and mCherry (Supplementary Fig. 15c). Expression of these probes did not change transferrin uptake without AP21967 treatment (Supplementary Fig. 17b). Consistent with this finding, the amount of surface transferrin receptors was not changed by the expression of FKBP/FRB probes (Supplementary Fig. 17c, d). We found that transferrin uptake decreased with AP21967 for PH–mCherry-FRB×2 but not for the control mCherry-FRB×2 (Fig. 5g). Furthermore, for the C-terminus-attached FKBP control (CLC-FKBP-EGFP), PH-mCherry-FRB×2 did not alter transferrin uptake (Fig. 5g). These results indicate that manipulation of the CLC N-terminal position inhibited transferrin uptake and endocytosis. These data are consistent with the decrease in the residence time of CCSs we observed in live cell imaging. From these collective data, we concluded that the conformational changes in CLC we mapped in FRET-CLEM experiments are important structural changes required for endocytosis of cargo-loaded vesicles in mammalian cells. Thus, the movement of the N-terminal domain away from the clathrin vertex and plasma membrane is a key regulatory step in clathrin-mediated endocytosis.

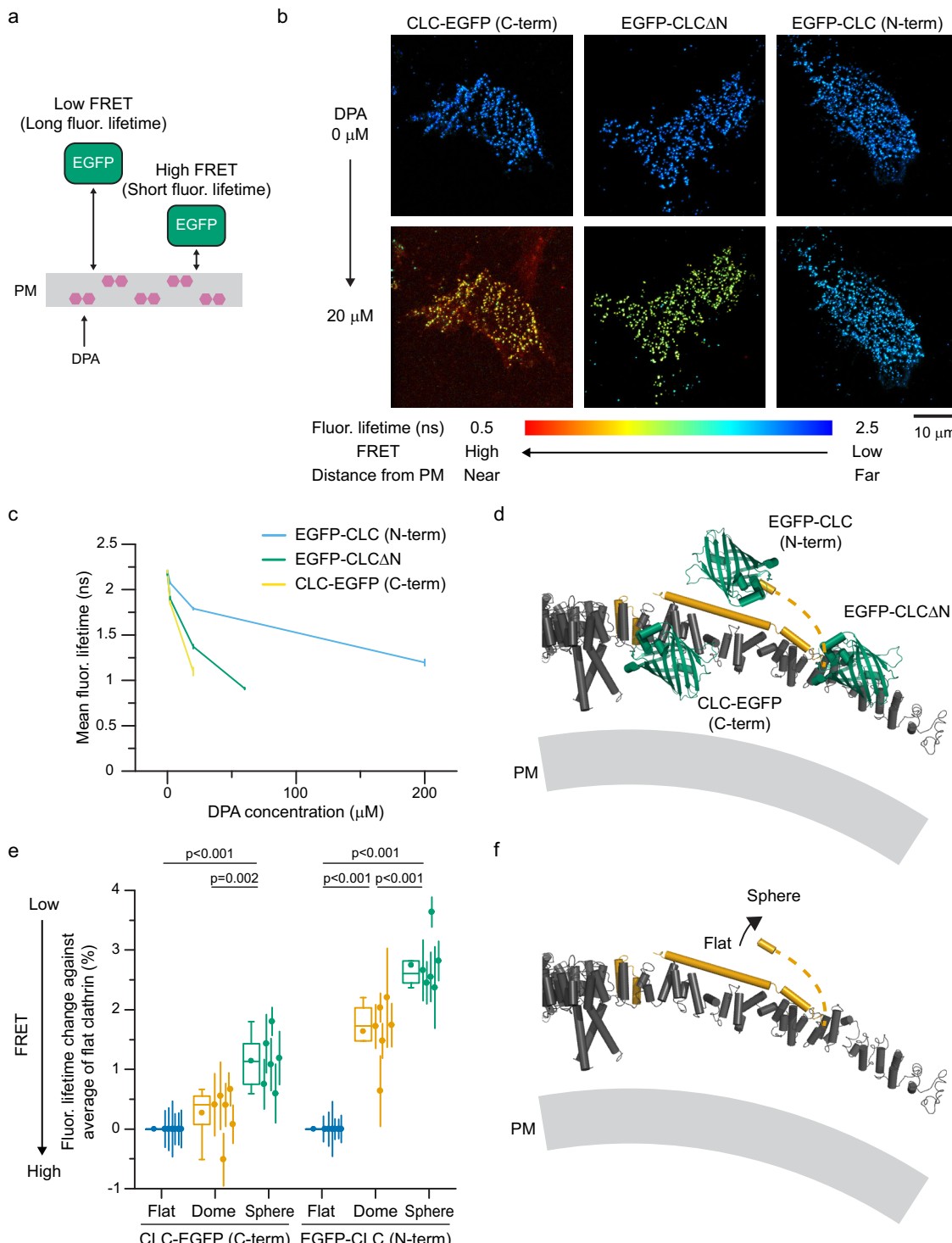

**Fig. 3 | CLC N-terminal position perpendicular to the plane of the plasma membrane. a** Dipicrylamine (DPA) is a nonfluorescent hydrophobic anion that incorporates into membranes. DPA quenches EGFP in a distance dependent manner by FRET. PM is the plasma membrane. **b** FLIM images of unroofed membranes of HeLa cells expressing CLC-EGFP (left), EGFP-CLCΔN (center), or EGFP-CLC (right) without (top) or with 20 μM DPA (bottom). Scale 10 μm. **c** Mean fluorescence lifetimes with different DPA concentrations. $n = 16$ (EGFP-CLC), 17 (EGFP-CLCΔN), and 15 cells (CLC-EGFP) from 3 experiments. Errors are SE. **d** A structural model of EGFP positions of CLC probes predicted from the EGFP-DPA FRET experiments. The model is based on PDB 4KW4 and 3LVG. **e** FRET-CLEM on HeLa cells expressing either CLC-EGFP or EGFP-CLC with DPA. DPA concentrations were 3 μM for CLC-EGFP and 80 μM for EGFP-CLC to obtain ~50% FRET efficiencies to make the degree

of fluorescence lifetime changes similar. Mean fluorescence lifetimes from single CCSs were analyzed by categorizing them according to lattice structures and compared to average values of flat structures. $n = 6$ cells from 4 experiments for each condition. Each dot is from one cell experiment and errors are SE. One-way ANOVA, then Tukey's test. For box plots, box is interquartile range, center line is median, center circle is mean, whiskers are minimum and maximum data points with a coefficient value of 1.5. **f** A proposed structural model of CLC conformational changes predicted from both FRET-CLEM with EGFP-ShadowY and EGFP-DPA. The N-terminus of CLC moves away from both the CLC C-terminus (triskelion vertex) and the plane of the plasma membrane as clathrin lattices curve. The structural model is based on PDB 3LVG. Source data are provided as a Source Data file.

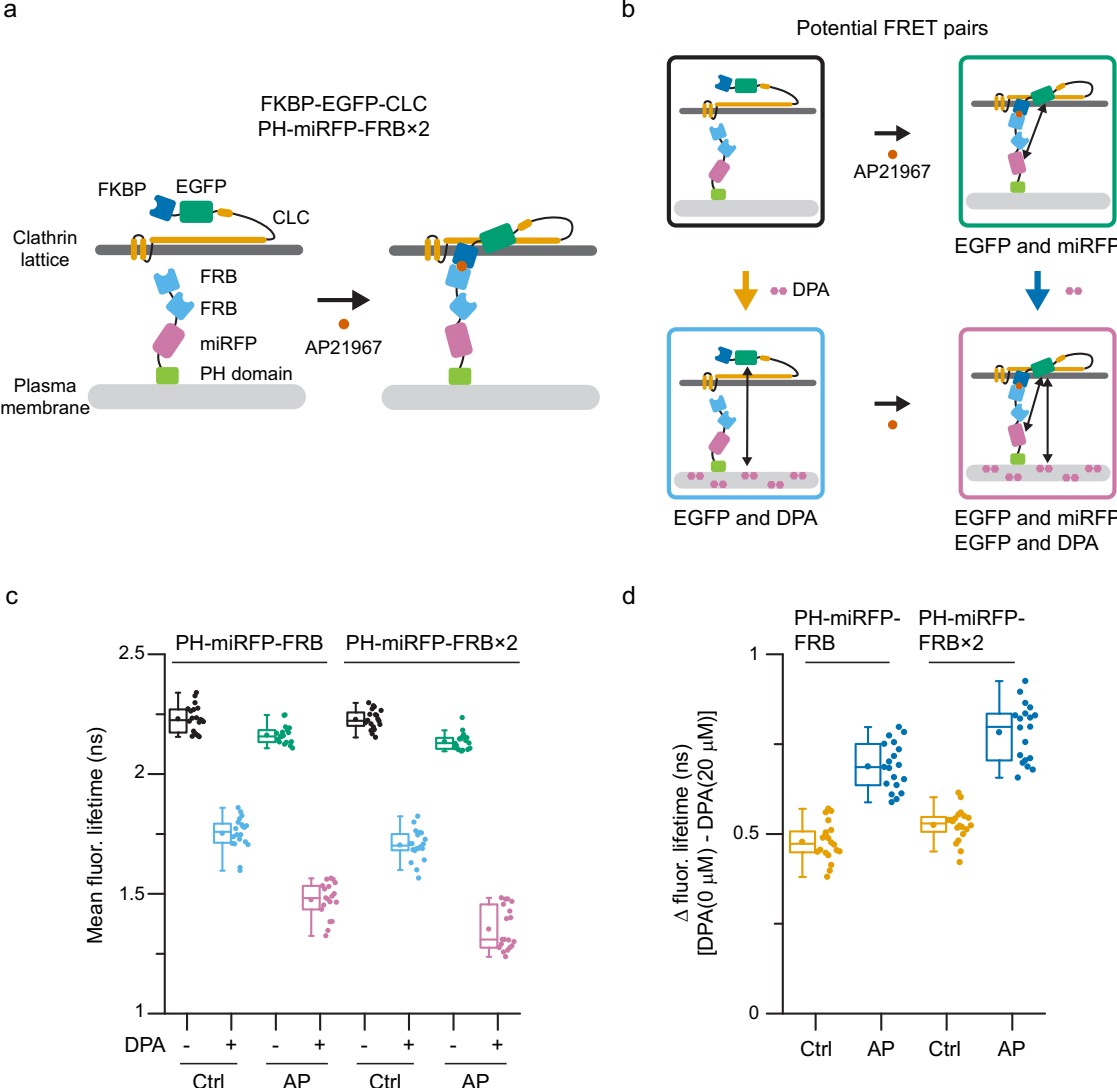

**Fig. 4 | Manipulation of CLC N-terminal position using a chemically inducible dimerization system. a** Schematic models of the chemically inducible FKBP/FRB dimerization system. FKBP is attached to the N-terminus of CLC, and two FRBs (FRB×2) are attached to the C-terminus of PH domain from PLCδ1. A rapamycin analog, AP21967, induces heterodimerization between FKBP and the T2098L mutant of FRB. **b** Potential FRET pairs without or with AP21967 treatment either absence or presence with DPA. **c** Fluorescence lifetime measurements were performed on unroofed membranes of HeLa cells expressing FKBP-EGFP-CLC either with PH-miRFP-FRB, or PH-miRFP-FRB×2 without or with 20 µM DPA. Cells were unroofed after 15 min incubation with AP21967 or ethanol (control). *n* = 20 (PH-miRFP-FRB, control), 19 (PH-miRFP-FRB, AP21967), 20 (PH-miRFP-FRB×2, control), and 19 cells (PH-miRFP-FRB×2, AP21967) from 3 experiments. **d** Differences in fluorescence lifetimes (shown in panel c) without and with DPA. For box plots, box is interquartile range, center line is median, center circle is mean, whiskers are minimum and maximum data points with a coefficient value of 1.5. Source data are provided as a Source Data file.

## Discussion

We have explored the molecular-scale conformational changes in clathrin light chain at clathrin sites at the plasma membrane of mammalian cells. We find that the N-terminus of CLC moves away from both the triskelion vertex and the plasma membranes as clathrin lattices curve. Blocking this movement increased clathrin lattice sizes at the plasma membrane and reduced transferrin endocytosis. Thus, specific structural changes in CLC in clathrin lattices at the plasma membrane are regulators of curvature and endocytosis in living mammalian cells.

Conformational changes in CLC have been proposed to regulate clathrin assembly[17]. We mapped the specific changes in cells by measuring vectors in two orthogonal axes. First, EGFP-DPA FRET measurements positioned the light chain relative to the plane of the membrane (Fig. 3d), and reported the movement along the axis perpendicular to the plasma membrane (Fig. 3f). Unlike classic in vitro models[17], we find that the N-terminus of CLC is located farther

into the cytoplasm than the proximal leg of clathrin heavy chain and moves away from the membrane. With intermolecular FRET between EGFP and ShadowY at the N- and C-terminus of the light chain (Fig. 2), we could map structures along the axis parallel to the plasma membrane. Here again, we detected a displacement of the N-terminal domain away from the CLC C-terminus (clathrin lattice vertex). Combined, these data indicate that the N-terminus of CLC moves away from both the clathrin vertex and the plane of the plasma membrane as lattices gain curvature.

Although the exact position of the N-terminus in relation to the heavy chain proximal leg domain was not determined, our experiments could be used to provide additional insights. Specifically, we modeled how FRET efficiencies between EGFP-CLC/CLC-ShadowY (Supplementary Fig. 5b) or EGFP-CLC/ShadowY-CLC (Supplementary Fig. 5g) would change as the N-terminal position changes according to the x-ray structures[17] across all possible spatial positions

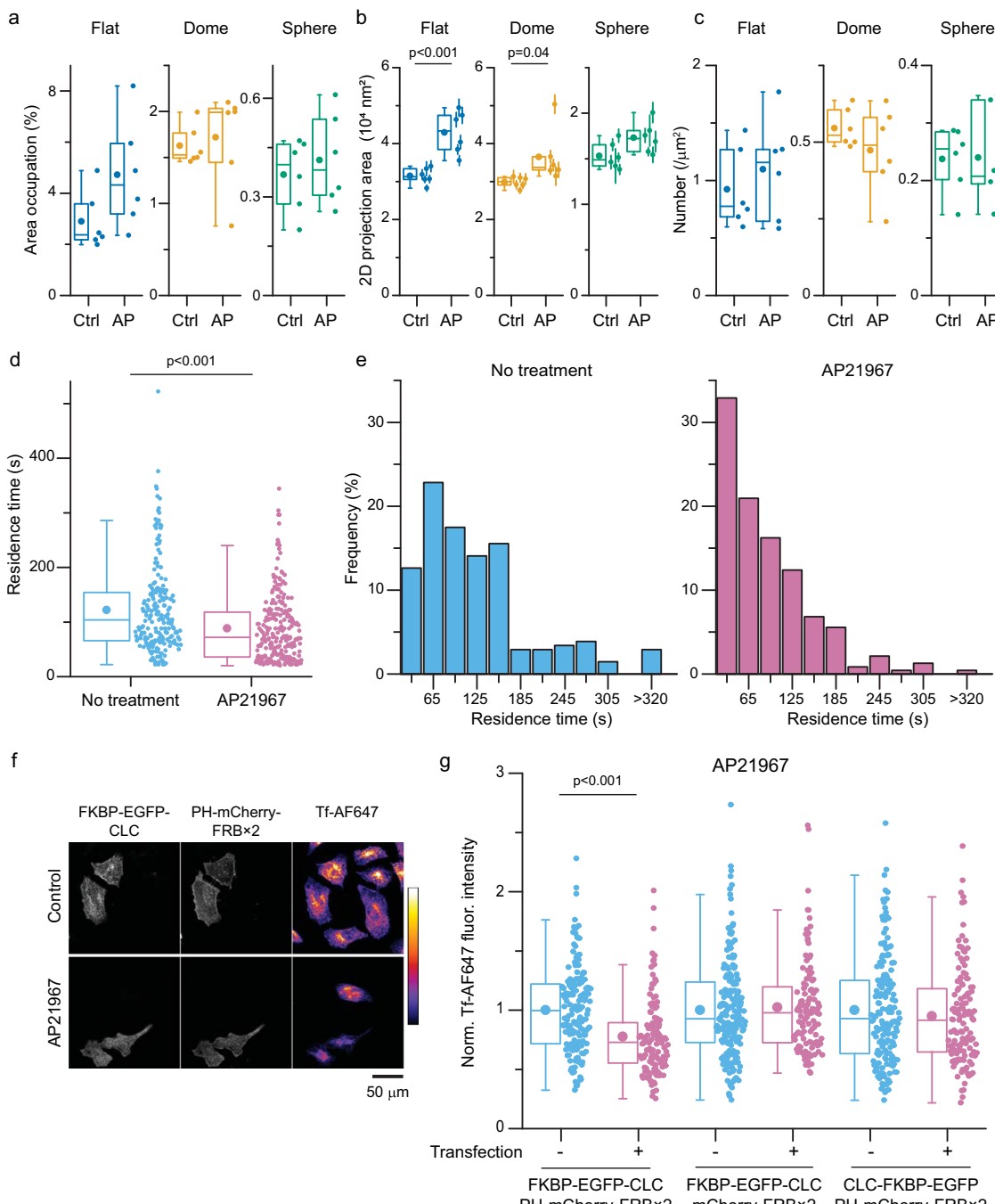

**Fig. 5 | Manipulation of CLC conformation changed lattice structures, dynamics, and endocytosis. a–c** Unroofed membranes from cells expressing FKBP-EGFP-CLC and PH-miRFP-FRB×2 treated with AP21967 (AP) or ethanol (control) were imaged with PREM. Two-dimension area of single CCS were manually segmented and measured. Membrane area occupation against the total analyzed membrane area (**a**), two-dimension projection area (**b**), and density (**c**) of flat, domed, and sphere CCSs were compared. Each dot is from one cell experiment, and errors are SE. *n* = 6 cells from 3 experiments for each condition. The average measured area / cell (mean ± SE) = 213 ± 34 (control) and 165 ± 10 μm² (AP21967). **d** Live cell time lapse TIRF imaging on HeLa cells expressing FKBP-EGFP-CLC and PH-miRFP-FRB×2 without or with AP21967 treatment. Tracks with over 20 s were analyzed and residence times were compared. *n* = 206 spots from 5 cells from 3 experiments (no treatment) and 234 spots from 5 cells from 4 experiments

(AP21967). A two-sided unpaired t test was used. **e** Histogram of residence times. **f** Confocal projection images of Alexa Fluor 647 conjugated transferrin (Tf-AF647) uptake in HeLa cells expressing FKBP-EGFP-CLC and PH-mCherry-FRB×2 treated with AP21967 or ethanol (control). Fluorescence intensity of Tf-AF647 (arbitrary units) is represented by pseudo color. *n* = 3 experiments. Scale 50 μm. **g** Transferrin uptake in HeLa cells expressing FKBP and FRB probes treated with AP21967. Fluorescence intensities of incorporated Tf-AF647 normalized by non-transfected cells in the same sample were compared between non-transfected and transfected cells. *n* = 116–198 cells from 3 experiments for each conditions. For box plots, box is interquartile range, center line is median, center circle is mean, whiskers are minimum and maximum data points with a coefficient value of 1.5. A two-sided unpaired t test was used. Source data are provided as a Source Data file.

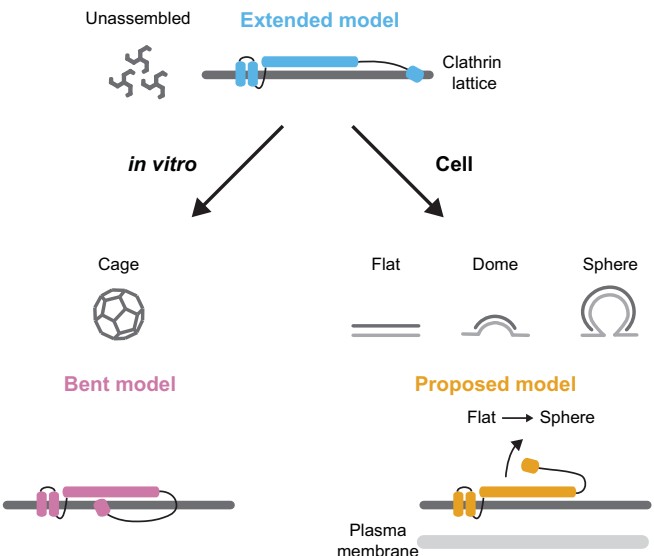

**Fig. 6 | Models of conformational switch in CLC in living cells.** Schematic models of conformational switch in CLC at the plasma membrane in cells. CLC assumes an extended conformation in unassembled triskelia in the cytoplasm similar to that seen in x-ray crystal structures. Next, when CLC assembles at the membrane as a flat lattice, CLC changes conformations from the extended to a new folded conformation. The N-terminus is then displaced deeper into the cytosol as clathrin lattices curve into vesicles. The extended and bent models are based on PDB 3LVG.

(Supplementary Fig. 18a). Although the fraction of heavy chains binding to ShadowY-attached CLC might affect the overall gross efficiency of FRET, the positional dependency and direction of change would not change with expression levels according to our calculations. Because FRET efficiencies were largely similar for both sites, the N-terminus of CLC is predicted to rest in the overlap regions that match these paired distances (Supplementary Fig. 18b, magenta circles). To accommodate these constraints, CLC would assume a slightly folded conformation (orange and green in Supplementary Fig. 18c) rather than a stretched conformation (blue in Supplementary Fig. 18c). From these considerations we propose a new model of CLC conformational changes (Fig. 6). Here, CLC assumes an extended and tightly bound conformation in unassembled triskelia in the cytoplasm similar to that seen in x-ray crystal structures and measured with FRET[17]. Next, when CLC assembles at the membrane as a flat lattice, the light chain changes conformations from this extended position into a new folded conformation. The N-terminus is then displaced deeper into the cytosol as clathrin lattices curve into vesicles. In support of our data, the average position of the CLC N-terminal domain is not visible by cryo-EM in purified clathrin-coated vesicles[46]. This suggests that the N-terminal domain is flexible and positioned away from the lattice in highly curved structures.

What initiates this structural change in the light chain? HIP and HIP1R interact with the CLC N-terminal residues[54,55]. Although CLC QQN mutants are thought to be deficient in binding HIP1 and HIP1R[55], we found that FRET between the N- and C- terminus of the CLC QQN mutants showed FRET changes comparable to wild type proteins (Fig. 2d, f). Thus, if this region is necessary for binding, HIP1R interactions are unlikely to drive these conformational changes. The binding of other proteins to the lattice[16,24], changes in protein-protein interactions caused by cargo loading[1], or phosphorylation[21] are alternate mechanisms that could drive this change. Finally, physical mechanisms such as crowding and cargo loading into the vesicle could induce the conformational switch[2,13].

In vitro studies have shown that CLC increases the rigidity of clathrin lattices on a solid substrate[20]. These effects were different between different CLC isoforms[23]. In living cells, CLC has been shown

to be important for endocytosis under conditions of high membrane tension[19,22]. Here, our FKBP/FRB experiments demonstrate that conformational changes in CLC regulate the structure of clathin (Fig. 5a–c), the dynamics of single clathrin sites in live cells (Fig. 5d, e), and the endocytosis of cargo in whole cells (Fig. 5f, g). One possible interpretation is that the lattice can grow but not fully curve due to the inhibition of the CLC conformation switch. Possibly, clathrin lattices grow larger because treatment disturbed the balance between lattice assembly and curvature[12]. These irregularly-assembled CCSs are then disassembled by proofreading mechanisms[3]. The CLCs role in endocytosis differs between isoforms, cargos, and cell types[19,21–23,56–59] and the physiological function of CLC are not fully understood[16,24]. Clarification on how CLC conformations are regulated across isoforms and cargos will provide a unified view of the light chain's mechanistic roles across different cells, tissues, and states.

Our work has some specific limitations. For example, we expressed FP-tagged probes to measure FRET. Thus, the expression ratio between endogenous and FP-labeled proteins could affect the overall FRET efficiency (Supplementary Figs. 4c and 18a). This, along with other photophysical issues, makes it difficult to convert FRET efficiencies to absolute atomic distances[60]. However, the differences in expression level do not affect the direction of FRET changes caused by conformational changes in this system (Supplementary Figs. 4c, 12, and 18a). Likewise, the fact that our two orthogonal FRET-based experiments using either FRET between two fluorescent proteins or FRET between a fluorescent protein and a probe in the membrane all showed similar vectors of movement for the light chain strongly supports our structural models. In the future, dual-tagging all alleles of the endogenous protein is a direction to further refine these measurements[31]. Also it will be interesting to investigate the relationship between the stoichiometry of proteins and FRET with a combination of other optical techniques. Furthermore, the size of fluorescent proteins, while relatively small compared to the size of the clathrin complex and approaching the size of large red organic dyes, may affect a protein's structure. Likewise, larger probes can limit the ability to detect small or distant conformational changes[60]. Thus, the incorporation of smaller tags or artificial amino acids are future directions to improve these measurements[61,62]. Indeed, smaller probes have been shown to better recapitulate absolute distance changes seen during conformational transitions and might have less of an influence on the proteins themselves[29]. In addition, increasing photon counts with organic dyes, combined with new quantitative analysis of photon count data, will be important for improving this method. In the case of clathrin light chain, however, fusion of fluorescent proteins has not been seen to perturb endocytosis or change clathrin lattice structures[63]. Expression of CLC probes used in FRET experiments did not change the size and number of clathrin lattice structures (Supplementary Fig. 19).

FRET efficiencies are determined by photophysical properties (excitation, emission, absorbance), distance, and orientation of the probes[28,44]. Here, we used a flexible linker to connect the FPs to CLC (Supplementary Fig. 6 and Methods). Thus, it is reasonable to propose that the population of FPs in a CCS can assume many possible orientations and the effect of an orientation bias would be small. However, we cannot exclude the small possibility that the curvature of clathrin lattices causes slight differences in the orientations between the donor and acceptor, which may affect the overall FRET efficiencies. However, the fact that the N-terminal changes was observed in both two FP-FRET experiments and FP-DPA FRET experiments, again supports the idea that orientation effects are not large or dominant. Combined, these caveats make it difficult to directly equate FRET efficiencies to atomic distances. This is commonly the case in FRET studies done within cells[60].

For cells expressing only EGFP probes, although there were no substantial change in fluorescence lifetimes among different lattice structures, clathrin vesicles tended to show slightly higher lifetimes

than flat clathrin structures (Fig. 2). Because Homo-FRET is not thought to change the overall fluorescence lifetime[44], this might be caused by a difference in the local environment[64], or a systematic counting error due to the differences in the photon count rate among CCSs with different fluorophore densities[65]. These effects, however, were small and consistent across samples.

Here, we developed a new correlative FLIM-FRET and PREM method to track the conformational changes in CLC at single sites of endocytosis in cells. The N-terminus of CLC makes a dramatic movement away from the clathrin lattice vertex and deeper into the cytosol as lattices curve. These conformational dynamics are key for clathrin-mediated endocytosis. These data, combined with the rich biochemical, functional, genetic, and biophysical information on membrane traffic, will lead to a more robust understanding of how endocytic proteins work together during clathrin-mediated endocytosis to drive the internalization of cargo. More generally, our new method maps molecular interactions and conformational changes of targeted proteins at identified sites in the complex environment of the cell. Thus, FRET-CLEM can be used to investigate conformational changes of any accessible membrane-associated protein including ion channels, transporters, receptors, adhesion proteins, or enzymes, in the context of their local plasma membrane environments[29,32].

## Methods

### Cell culture

HeLa cells (ATCC, CCL-2) were maintained at 37 °C, with 5% $CO_2$ in DMEM (Gibco, 11995073) supplemented with 10% fetal bovine serum (ATLANTA biological, S12450H) and 1% vol/vol penicillin/streptomycin (Invitrogen, 15070-063). SK-MEL-2 cells (ATCC, HTB-68) were maintained at 37 °C, with 5% $CO_2$ in EMEM (ATCC, 30-2003) supplemented with 10% fetal bovine serum and 1% vol/vol penicillin/streptomycin. The cells were placed on poly-L-lysine coated coverslips (Neuvitro, GG-25-1.5-PLL). They were transfected with 0.75 mL of Opti-MEM (Life Technologies, 31985062), 3.8 μL Lipofectamine 2000 (Life Technologies, 11668027), and 1.5 μg of DNA for 3 h after being introduced to the cells. Then, transfected cells were incubated in DMEM or EMEM growth medium for 20–30 h before experiments. The cells tested negative for mycoplasma contamination.

### Plasmids

ShadowY (#104621), mScarlet-I (#85068), mTurquoise2 (#54842), miRFP703-CLCb (#79997), CFP-FKBP (#20160), Nup54-EGFP494(0) (#163426), Nup54-EGFP494(1) (#163427), Nup54-EGFP494(flex0) (#163429), and Nup54-EGFP494(flex1) (#163430) were purchased from Addgene, and pSNAP-tag(m) was purchased from New England Biolabs. CLCa-GFP was kindly donated by Dr. W. Almer (Oregon Health & Science University). Lyn-GFP-FRB was kindly donated by Dr. T. Inoue (Johns Hopkins University). mNeonGreen was kindly donated by Dr. J. Shah (Harvard University). mCherry-PH was from our previous studies[66]. All plasmids used in this study were constructed using either Q5 Site Directed Mutagenesis Kit (New England Biolabs, E0554S) or In-Fusion HD Cloning Plus (Clonetech, 638920) following manufacturer's instructions. For tandemly-connected FPs in Supplementary Fig. 2a, N- (residues 1–4 for ShadowY, and 1–5 for mCherry and mScarlet-I) and C-terminal disordered regions (residues 228–238 for EGFP and mTurquoise2, and 226–236 for mNeonGreen) of FPs were deleted. And two FPs were connected with a short linker (GGSGGS). For EGFP-ShadowY-CLC in Fig. 1, tandemly-connected EGFP-ShadowY was fused to CLCa with a 10 a.a. linker. For CLCb probes used in FRET experiments, either C-terminal disordered region (residues 228–238) deleted or N-terminal disordered region (residues 1–4) deleted EGFP (or ShadowY) was connected to CLCb with a short linker (GGSGGS). All plasmids were confirmed by sequencing (Psomagen) and identified as in Supplementary Data 1.

### Fixing and unroofing

Cells were rinsed in intracellular buffer (70 mM KCl, 30 mM HEPES maintained at pH 7.4 with KOH, 5 mM $MgCl_2$, 3 mM EGTA), and manually unroofed with 19-gauge needle and syringe using 2% paraformaldehyde (Electron Microscopy Sciences, 15710) in the intracellular buffer. After unroofing, the coverslips were transferred to fresh 2% paraformaldehyde in the intracellular buffer for 20 min. They then washed with phosphate-buffered saline (PBS). Imaging was performed in PBS (pH 7.4) (Quality Biological, 114-058-131).

For FKBP/FRB dimerization experiments, cells were unroofed after 15 min incubation with 500 nM AP21967 (500 μM stock in ethanol) (Takara, 635055) or 0.1% w/v ethanol (control) in DMEM growth medium.

### Imaging of unroofed plasma membranes for CLEM

Unroofed cells were stained either with 16.5 pmol of Alexa Fluor 350-phalloidin (Life Technologies, A22281), Alexa Fluor 568-phalloidin (Life Technologies, A12380), or Alexa Fluor 647-phalloidin (Life Technologies, A22287) for 15 min depending on spectra of expressing FPs. Then cells were rinsed with PBS. 1 mm × 1 mm large montage was generated for proteins of interest and phalloidin using a Nikon Eclipse Ti inverted microscope with a 100×, 1.49 NA objective (Nikon, SR HP Apo TIRF) and an Andor iXon Ultra 897 EM-CCD camera under the control of Nikon Elements software. Images were obtained by TIRF illumination except for Alexa Fluor 350 which was imaged by epi illumination. This map was used to find the cells expressing the target proteins for FLIM and CLEM analysis. The imaged area was marked with a circle (4 mm in diameter) around the center of the imaged area using an objective diamond scriber (Leica, 11505059)[40,67]. The immersion oil was carefully removed from the bottom of the glass coverslip. The sample was subsequently imaged by FLIM or stored in 2% glutaraldehyde (Electron Microscopy Sciences, 16019) at 4 °C until EM sample preparation.

### FLIM

Time-domain fluorescence lifetime imaging was performed with a Leica Falcon SP8 confocal microscope with a 63×, 1.40 NA oil immersion objective (Leica, HC PL APO CS2) under the control of Leica LAS X software. As first, a large image montage around the region of interest on the coverslip marked by a diamond scriber was acquired to find the cells which were identified in the TIRF montage. Then, FLIM images were collected at a lateral spatial resolution of 80 nm per pixel and with a scan speed at 3.16 μs per pixel. The confocal aperture was set at a diameter of 168 μm (2 AU @510 nm). EGFP was excited by 488 nm and 498–560 nm fluorescence was collected. A notch filter for 488 nm was used. To reduce the impact of pile-up effect, an excitation power was set to keep peak count per pulse as less than 0.1. For CLEM, the imaged sample was then stored in 2% glutaraldehyde at 4 °C until EM sample preparation.

To select a FP pair for FLIM-FRET, we made tandemly connected FPs and compared their FRET efficiency (Supplementary Fig. 2a). Based on its high FRET efficiency and compatibility for our optical systems, we selected monomeric EGFP and ShadowY (Supplementary Fig. 2b). To check the relationship between the number of frames and photobleaching or FRET efficiency change, fluorescence intensity and FRET efficiency of EGFP-ShadowY-CLC was compared with various frame numbers (Supplementary Fig. 3e,f). Fluorescence intensity did not change and FRET efficiency slightly decreased as the frame number increases. This might be due to photobleaching of ShadowY occurs faster than EGFP. Although the magnitude of ShadowY photobleaching should be smaller for lower FRET efficiency situation like intermolecular FRET, we used 150 frames to minimize the effect of photobleaching. With 150 frames, ~10,000 photon counts per single CCS were obtained in average (Supplementary Fig. 3g). With this range of photon counts, FRET efficiency ($E$) of EGFP-ShadowY-CLC from single CCS could be

estimated by fitting a fluorescence decay curve with a bi-exponential function convolved with the Gaussian pulse response function[68] (Supplementary Fig. 3h).

$$F(t) = F_0[P_1 H(t, t_0, \tau_1, \tau_G) + (1 - P_1) H(t, t_0, \tau_2, \tau_G)] \quad (1)$$

$$H(t, t_0, \tau_1, \tau_G) = \frac{1}{2} \exp\left(\frac{\tau_G{}^2}{2\tau_1} - \frac{t - t_0}{\tau_1}\right) \text{erfc}\left(\frac{\tau_G{}^2 - \tau_1(t - t_0)}{\sqrt{2}\tau_1\tau_G}\right) \quad (2)$$

where $F(t)$ is the fluorescence lifetime decay curve, $F_O$ is the peak fluorescence before convolution, $P_1$ is the fraction of the first component, $\tau_1$ and $\tau_2$ are the fluorescence lifetime of first and second components, $\tau_G$ is the width of the Gaussian pulse response function, $t_O$ is time offset, and erfc is the complementary error function.

$$\tau_a = P_1 \tau_1 + (1 - P_1)\tau_2 \quad (3)$$

$$E = 1 - \frac{\tau_{a,DA}}{\tau_{a,D}} \quad (4)$$

where $\tau_a$ is amplitude-weighted fluorescence lifetime, $\tau_{a,D}$ is amplitude-weighted fluorescence lifetime of the donor without acceptor, and $\tau_{a,DA}$ is amplitude-weighted fluorescence lifetime of the donor in presence of the acceptor[44]. Values from single CCSs (Supplementary Fig. 3h) were consistent with those from whole plasma membrane (Supplementary Fig. 3a–c). However, in general, nearly 10,000 photon counts are required to fit a bi-exponential[45,69]. So, curve-fitting on the fluorescence decay curves from very small CCSs will not be adequate. Thus, instead to estimate FRET efficiency through curve-fitting, mean fluorescence lifetime, the center mass of the fluorescence lifetime decay, was used as an indicator for FRET efficiency (Fig. 1d and Supplementary Fig. 3d). Background was not subtracted. In our measurements, background counts were generally very low and signal-to-background ratio was >100 in most experiments (Supplementary Fig. 8b). Mean fluorescence lifetime was not dependent on the signal-to-background ratio.

For FLIM measurements with dipicrylamine (DPA; City Chemical LLC), a 20 mM stock solution of DPA in DMSO (Sigma-Aldrich, D2650) was prepared fresh from powder every day and diluted to a final concentration with PBS. Imaging was performed at least 10 min after addition of DPA. FRET efficiency between DPA and miRFP should be very low because DPA absorbance spectra and miRFP emission spectra are sufficiently separated (Supplementary Fig. 2c).

For live cell FLIM imaging (Supplementary Fig. 11), cells were imaged in imaging buffer (130 mM NaCl, 2.8 mM KCl, 5 mM CaCl₂, 1 mM MgCl₂, 10 mM HEPES, and 10 mM glucose at pH 7.4) at 21 °C.

## Platinum replica EM
EM samples were prepared as described previously[4,67]. Coverslips were transferred from glutaraldehyde into 0.1% w/v tannic acid for 20 min. Then, they were rinsed 4 times with water, and placed in 0.1% w/v uranyl acetate for 20 min. The coverslips were then dehydrated, critical point dried with a critical point dryer (Tousimis Samdri, 795), and coated with platinum and carbon with a freeze fracture device (Leica, EM ACE 900). The region of interest on the coverslip marked by a diamond scriber was imaged with a 20× phase-contrast objective to obtain another map of the region imaged in fluorescence. The replicas were lifted and placed onto formvar/carbon-coated 75-mesh copper TEM grids (Ted Pella, 01802-F) that were freshly glow-discharged with PELCO easiGlow 91000. Again, the grid was imaged with a 20× phase-contrast objective to find the same region that was originally imaged in fluorescence. Each cell of interest was located on the grid prior to EM imaging[40]. TEM imaging was performed as previously described[37] at ×15,000 magnification (1.2 nm per pixel) using a JEOL 1400 and

SerialEM freeware for montaging[70]. Electron microscopy montages were processed using IMOD freeware[71].

## FRET-CLEM image analysis
The FLIM images were aligned to the EM images using an affine spatial transformation with nearest-neighbor interpolation to map the CCSs visible in both FLIM images and EM images[36,41]. Since position of sphere clathrin is sometimes changed during critical point drying due to a weak attachment to the membrane, we used flat and domed clathrin as fiducials[67]. Rectangular ROI was created around single isolated CCS, and mean fluorescence lifetime within each ROI was calculated. These values were analyzed by categorizing according to lattice structures, and were compared to the average value of flat clathrin for each unroofed membrane. This is because FRET efficiencies, and thus fluorescence lifetimes, varied among cells due to the differences in expression levels of donor and acceptor probes, or incorporated density of DPA (Supplementary Figs. 5, 9, 10, 13, and 14).

For FRET between EGFP and ShadowY, because ShadowY expression cannot be confirmed with fluorescence, only cells with a cellular average fluorescence lifetime of less than 2.1 ns were analyzed to ensure the expression of ShadowY.

## EM image analysis
Binary masks of the flat (no visible curvature), domed (curved but can still see the edge of the lattice), and sphere clathrin (curved beyond a hemisphere such that the edge of the lattice is no longer visible) were manually segmented (Supplementary Fig. 19a)[43,72]. The percentage of occupied membrane area was defined as the sum of areas from clathrin lattices of the specified subtype divided by the total area of measured membrane. The expression of CLC probes did not change the size and density of clathrin lattice structures (Supplementary Fig. 19b–d).

## Live cell TIRF imaging and analysis
Images were acquired using a Nikon Eclipse Ti inverted microscope with a 100×, 1.49 NA objective and an Andor iXon Ultra 897 EM-CCD camera. HeLa cells in the phenol red free DMEM growth medium [DMEM (Gibco, 31053036) with 10% fetal bovine serum, 1:100 dilution of 100× GlutaMAX (Gibco, 35050061), 1:100 dilution of 100 mM sodium pyruvate (Gibco, 11360070)] were mounted in a chamber at 37 °C with a water bath and continuous flow of humidified 5% CO₂ to maintain the osmolality and pH of the medium. The pixel size was 110 nm. TIRF images were acquired with 100 ms exposures at 0.5 Hz for 10 min. For AP21967 treatment, images were acquired at least 10 min after the addition of 500 nM AP21967.

FKBP-EGFP-CLC spots were tracked using the ImageJ plugin TrackMate[73]. Spot detection was done with the difference of Gaussians approach, and tracking was done using the simple linear assignment problem (LAP) tracker algorithm with a linking maximum distance of 330 nm per frame. Tracks that appeared and disappeared during the whole movie with over 20 s duration, and a total net displacement was less than 330 nm were selected for residence time analysis. Further, each track was visually inspected for isolation and tracking errors.

## Transferrin uptake assay
HeLa cells were incubated in starvation medium [DMEM containing 20 mM HEPES at pH 7.4 and 0.1% w/v bovine serum albumin (Fisher Bioreagents, BP9703)] for 1 h in an CO₂ incubator. They were then incubated with 500 nM AP21967 or 0.1% w/v ethanol (control) in starvation medium for 5 min. Then, 25 mg/mL Alexa Fluor 647 conjugated human transferrin (Invitrogen, T23366) was added to the starvation medium[74]. After 15 min incubation, the cells were fixed with 2% paraformaldehyde at room temperature for 25 min. They then washed with PBS. The cells were imaged with a Leica Falcon SP8 confocal microscope with a 63×, 1.40 NA oil immersion objective. Images were collected at a lateral spatial resolution of 120 nm per pixel. The

confocal aperture was set at a diameter of 191 μm, and optical sections with z-spacing of 0.8 μm were collected. Alexa Fluor 647 was excited by 633 nm and 638–800 nm fluorescence was collected.

The stack images were recombined using a sum-intensity operation. And background intensity from non-cell regions were subtracted. Then, cell outlines were traced manually and the mean intensity (total intensity/area) was measured for individual cells. Fluorescence intensity was normalized by the average value of non-transfected cells for each coverslip.

## Immunocytochemistry

HeLa cells were fixed in 2% paraformaldehyde in PBS for 25 min, blocked with 3% w/v bovine serum albumin for 60 min, and reacted with mouse monoclonal antibody against transferrin receptor (Santa Cruz, sc-65877, 1:50). The primary antibody was visualized by secondary antibody staining using goat anti-mouse IgG conjugated to Alexa Fluor 647 (Invitrogen, A21237, 1:500). The cells were imaged with a Leica Falcon SP8 confocal microscope with a ×63, 1.40 NA oil immersion objective. Images were collected at a lateral spatial resolution of 120 nm per pixel. The confocal aperture was set at a diameter of 191 μm, and optical sections with z-spacing of 0.8 μm were collected. Alexa Fluor 647 was excited by 633 nm and 638–800 nm fluorescence was collected. Images were analyzed as transferrin uptake assay.

## Linker flexibility measurements with polarization-TIRF (pol-TIRF) microscopy

HeLa cells were imaged with 3i vector TIRF microscopy system, based on an inverted microscope (IX-81; Olympus) and equipped with 100×, 1.49 NA objective (Olympus). Fluorescence was excited by 488 nm and 561 nm lasers passed through LF405/488/561/635 filters (Semrock). Emitted light was then divided by the image splitter's dichroic (565DCXR) and projected side-by-side through 525Q/50 and 605Q/55 emission filters onto the chip of an EMCCD camera (Andor, DU 897). Images were acquired using SlideBook6 (3i). A sequence of 40 images was taken with pol-TIRF fields that were parallel (s-pol) or perpendicular (p-pol) to the coverslip. Each individual image had an exposure time of 100 ms. For calibration, fluorescein solution (Alfa Aesar, L13251) was measured. 20 s-pol and 20 p-pol images were summed and background was subtracted. A p:s ratio was then calculated and normalized by the value of fluorescein measurements.

Past work measured the orientation of nucleoporins (Nups) in the nuclear pore complex using pol-TIRF microscopy and orientational sensors[75]. In these sensors, a short N-terminal α-helix of EGFP is conjugated directly to a C-terminal α-helix in a protein of interest by a continuous α-helix. When the number of amino acids in this linker helix is varied, EGFP rotates around the linker helix axis by angles dictated by α-helical geometry. Since the excitation dipole is fixed within EGFP, the p:s ratio shifts by changing the linker length. On the other hand, if EGFP is conjugated by a flexible linker, the p:s ratio does not shift by changing the linker length. First, we tested if we could reproduce the previous results in our imaging system. We imaged the bottom of the nucleus of fixed HeLa cells expressing Nup54-EGFP494 constructs[75] (Supplementary Fig. 6a). Consistent with previous data, the p:s ratio shifted by changing the linker length for the rigid linker probes but did not shift for the flexible linker probes (Supplementary Fig. 6b). Next, we applied this strategy to investigate the linker flexibility in CLC probes. We conjugated EGFP to C-terminal α-helix domain of CLC (at residue 206) by rigid or flexible linkers (Supplementary Fig. 6c). We then imaged unroofed HeLa cells expressing CLC orientational sensors, CLC-EGFP, or EGFP-CLC (Supplementary Fig. 6d). The p:s ratio shifted with different lengths of the rigid linkers however the p:s ratio was not changed by varying lengths of the flexible linkers. The p:s ratio of CLC-EGFP and EGFP-CLC which were used in FRET experiments were similar to those of the flexible linker sensors. These results support the assumption that the linkers of CLC-EGFP and

EGFP-CLC are flexible. Further, since there were no significant correlation between EGFP intensity and the p:s ratio in CLC-EGFP and EGFP-CLC (Supplementary Fig. 6e–g), linkers are likely flexible in clathrin lattices with various shapes.

## Western blot

HeLa cells were dissociated with trypsin and collected. After washing and centrifuge, 100 μL RIPA lysis buffer (Millipore 20–188) supplemented with protease inhibitor cocktail (Halt, 78429) was added and samples were incubated for 1 h on ice. After vortex, cell lysates were centrifuged at $21300 \times g$ for 15 min at 4 °C. The supernatant was used for SDS-PAGE on 4–12% Bis-Tris gels (Invitrogen, NP0335) with Tris-buffered saline with 0.1% Tween-20 (TBS-T) at 150 V. Blotting was performed by iBlot (invitrogen) with nitrocellulose membranes (Invitrogen, IB301002) accordingly to the manufacturer's protocol. Afterwards, membranes were incubated for 2 h in 5% milk in TBS-T. Primary antibodies [anti-CLCa/b (Millipore, AB9884, 1:5000), anti-CLCa (Sigma, HPA050918, 1:1000), or anti-CLCb (Abnova, H00001212-M01, 1:500)] were diluted in 5% milk/TBS-T and applied on the membranes overnight. After 5 times 5 min washing with TBS-T, secondary antibodies [anti-mouse IgG-HRP (Jackson ImmunoResearch Labs, 115-035-174, 1:2000) or anti-rabbit IgG-HRP (Jackson ImmunoResearch Labs, 211-032-171, 1:2000)] were diluted in 5% milk/TBS-T and applied on the membranes for 1 h. After washing with TBS-T, the membrane was imaged with ChemiDoc system (Bio-Rad) with ECL solution (cytiva, RPN2232). Then after washing, the membrane was stained with anti-βactin-HRP (Cell Signaling, 5125 S, 1:2000) and imaged.

## Estimation of expression levels of transfected constructs

To estimate the expression levels of transfected constructs, we performed western blot (Supplementary Fig. 7). First, we stained with anti-CLCa/b antibody (Supplementary Fig. 7a). The amount of endogenous CLCa/b decreased by transfection (Supplementary Fig. 7d). The degree of decrease was lager in dual-transfection than single-transfection. The reactivity of this antibody against N-terminal-FP-attached CLC was weaker than that against C-terminal-FP-attached CLC. Thus we did not measure the ratio between transfected and endogenous CLC with this antibody. Next, we stained with either anti-CLCa (Supplementary Fig. 7b) or anti-CLCb (Supplementary Fig. 7c) specific antibody. Consistent with anti-CLCa/b antibody, the amount of endogenous CLCa and CLCb decreased with transfection (Supplementary Fig. 7e,f). And the amount of transfected CLCb was larger than that of endogenous CLCb (Supplementary Fig. 7g). From these results, we modeled the expression of transfected constructs. CLCa is the dominant isoform in HeLa cells[18]. Thus, we assumed a ratio between CLCa and CLCb of 4:1 and a transfection efficiency of 80%. Under these conditions, we estimate the CLC amount in transfected cells (Supplementary Fig. 7h). In this model, the total amount of CLCs increases 10 times after transfection and the ratio between transfected and endogenous CLCs is 25:1 for single transfection and 100:1 for double transfection. Thus, in our FRET experiments, we propose that most CLCs are attached to an FP.

## FRET simulation

FRET simulations (Supplementary Figs. 4c and 18) were performed using MATLAB. We used 60 Å for Förster radius for EGFP-ShadowY FRET. For the simulation in Supplementary Fig. 4c, we determined the position of N- and C-terminus of CLC according to the structure models (PDB 3LVG and 6WCJ)[17,46]. For the simulation in Supplementary Fig. 18, the lateral position of N-terminus was moved by 0.6 Å spacing, and N-terminus was assumed to locate axially 25 Å higher than C-terminus. When we focus on single clathrin heavy chain, there are four different states; without CLC, endogenous

CLCa or CLCb binding, EGFP-CLC binding, or CLC-ShadowY binding (Supplementary Fig. 4b). By considering these four states, we calculated the FRET efficiency between EGFP-CLC and CLC-ShadowY or ShadowY-CLC binding to surrounding five clathrin heavy chains at various heavy chain occupancy with ShadowY-attached CLC.

$$\text{Occupancy} = \frac{[\text{ShadowY probe binding}]}{[\text{No CLC binding}] + [\text{Endogenous CLCa/b biding}] + [\text{EGFP} - \text{CLC binding}] + [\text{ShadowY probe binding}]} \tag{5}$$

### Quantification and statistical analysis

Image analysis and quantification were performed with ImageJ[76] and MATLAB. The statistical tests used for each experiment and the exact sample numbers ($n$ values) are indicated in the corresponding figure legends. p values of <0.05 were considered statistically significant. All statistical analysis and fitting were performed using Origin 2016 (Origin Lab). The exact p values are provided in Source Data file.

### Materials availability

The plasmids used in the study are deposited at Addgene (Supplementary Data 1).

### Reporting summary

Further information on research design is available in the Nature Portfolio Reporting Summary linked to this article.

## Data availability

The data generated in this study has been deposited in Figshare at https://doi.org/10.25444/nhlbi.c.6259170. The remaining data are available in the Article or Supplementary Information files. Source data are provided with this paper. Atomic coordinates of previously determined X-ray or Cryo-EM structures are available in the PDB under the following accession codes: 3LVG (clathrin heavy chains and light chains), 6WCJ (clathrin-coated vesicle), and 4KW4 (GFP). Source data are provided with this paper.

## Code availability

MATLAB codes used in this study are specific to lab file formatting. The codes are available in Figshare at https://doi.org/10.25444/nhlbi.14502156.

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

## Acknowledgements

We thank Dr. L.D. Islas (Universidad Nacional Autónoma de México) and Dr. H. Murakoshi (National Institute for Physiological Sciences) for sharing the absorbance and emission spectra of DPA and ShadowY respectively. We thank Dr. J. Jiang (National Heart, Lung, and Blood Institute) for use of the ChemiDoc. We thank National Heart, Lung, and Blood Institute Light Microscopy Core and Electron Microscopy Core for use of instruments and advice. We thank G. Haber for help with coding. We thank members of the Taraska lab for helpful discussions and edits. J.W.T. is supported by the Intramural Research Program of the National Heart, Lung, and Blood Institute, National Institutes of Health. K.O. is supported by JSPS Research Fellowship for Japanese Biomedical and Behavioral Researchers at NIH.

## Author contributions

K.O., K.A.S., and J.W.T designed the research. K.O. performed experiments and analysis. K.A.S. developed software for analysis. M.-P.S. helped with molecular cloning and western blots. J.W.T. supervised the project. K.O. and J.W.T. wrote the paper. All authors contributed to the interpretation of the data and commented on the manuscript.

## Funding

## Competing interests

The authors declare no competing interests.
