## [Peer Review File · Nature Communications]

A conformational switch in clathrin light chain regulates lattice structure and endocytosis at the plasma membrane of mammalian cellsReviewers' Comments:

Reviewer #1:

Remarks to the Author:

In the manuscript entitled "A conformational switch in clathrin light chain regulates lattice structure and endocytosis at the plasma membrane of mammalian cells", Obashi and colleagues from the Taraska group develop a correlative fluorescence resonance energy transfer (FRET) and platinum replica electron microscopy method, named FRET-CLEM. They investigate the conformational changes in clathrin light chain by mapping distance changes both parallel and perpendicular to the plasma membrane. The authors find that the N-terminus of CLC moves away from the plasma membrane and triskelia vertex as lattices curve and that preventing this conformational switch increased clathrin structure size and inhibited endocytosis.

This work focuses on an important topic and overall, the experimental design/analysis is of high quality and the writing is good. From a methodological standpoint this work is original however the conclusions drawn are mitigated and not verified but other techniques leaving the possibility that the authors could be wrong. Also, in my opinion, the conclusions drawn are sometimes exaggerated and not controlled enough. For example, the schemes show that each one of the three light chains is tagged but in reality, the endogenous proteins occupy some of these positions and make the interpretations very different from the theoretical models drawn. Also, in the same line FRET results are difficult to interpret as expression levels of transfected constructs cannot be controlled. Overall, several problems detailed below will need to be addressed before publication in Nature communications:

Major comments (not listed in order of importance):

1/ One of the major claims in this work concerns the fact that the authors mention throughout the manuscript that there are three light chains for three heavy chains. Although this idea has been around in the field, it has been shown by the Blondeau group (Girard et al., 2005 MCP) using quantitative proteomics that this is not true in non-neuronal cells and that the clathrin heavy and light chains are expressed with a non-stoichiometric relationship. While this might seem trivial, in the particular work it is a major issue as it alters the interpretation of the FRET analysis.

2/ On line 182 the authors state "However, in these mutants, we found similar fluorescence lifetime changes as in the wild type protein (Fig 2D and f). These data do not support model 2". The authors should also take into consideration the presence of endogenous CLC contribution in these clathrin-coated structures.

3/ Lifetimes depend on the local environment and rigidity. Do expression levels of these constructs play a role in clathrin lattice formation? The authors need to provide the quantification of all these constructs expression levels by using for example Western blot experiments. Does the EGFP tagged C-terminal CLC construct show the same lifetimes?

4/ On line 99 the authors state that "We chose monomeric EGFP and a dark yellow fluorescent protein, ShadowY, as an optimized FRET pair by comparing six potential fluorescent protein (FP) pairs (Fig S2A)". I assume all these FRET Pairs (donor and acceptors) are separated by the same length linker? are all these FRET pairs tagged to CLC? In the figure legend, it mentioned tandemly connected, more details are needed.

-EGFP- ShadowY-CLC (Fig 1A) I did not find the information about this construct in the methods section, not sure how well GFP and Shadow Y are separated

5/ What happens to the lifetime if expressing both CLC-EGFP and EGFP-CLC? are we expect an

intermediate lifetime? Because stoichiometry might change, and homoFRET?

6/ In the last part of the results section, the authors investigated whether manipulation of the CLC N-terminal position effects transferrin endocytosis in whole cells describe using Transferrin uptake as a functional assay which allows to quantification the clathrin-mediated endocytosis. While this is an important experiment, the amount of transferrin receptors at the surface could be altered in cells transfected with the different constructs and not reflect the true endocytic activity. One should know the expression levels of transferrin receptors upon expressions of different tagged CLC The authors should control this by showing that the transferrin receptor levels are unchanged and that uptake truly reflects endocytosis activity.

7/ Graphical representations of EGFP CLC, Yshadow CLC, and EGFP-Yshadow CLC are a bit confusing to the reader in Fig1 and Fig.2. Besides graphical representations, 2c is misleading because one should show all scenarios and combinations of donor and acceptor CLC, inter FRET, HomoFRET, and stoichiometry.

8/On line 332, the authors state that "The roles of other proteins or phosphorylation are alternate mechanisms that could drive this change". What about the cargo effect? how does it change the CLC N-term configuration? Interactions with other proteins might lead to restriction of the dipole-dipole moment of the donor and acceptor, which affects the FRET. The authors need to provide the experimental evidence that shows protein-protein interactions does not influence the N-term conformational rearrangements.

9/ The present study focuses exclusively on HeLa cells. Although widely used, HeLa cells are cancerous with many intrinsic abnormalities and are notoriously known for forming large flat lattices. What about other cell types where various sizes of CCPs and structures are present in various cell types. It is very important to test this proposed model in other cell types where the clathrin behavior is different such as for instance BSC-1 cells which only form large pits or neuronal cells which form smaller ones.

10/As mentioned in the discussion progress from flat clathrin lattice to spherical clathrin structures leads to differences in the orientations between donor and acceptor, which majorly affects the lifetime of the donor, and results in alterations in FRET efficiency. Considering this without knowing the stoichiometry of CLC in clathrin-coated structures, lifetime based FRET efficiency leads to misinterpretations...

11/ On line 772, in the legend for FigS2: "FRET efficiencies of tandemly-connected fluorescent proteins (FPs). Fluorescence lifetimes were measured in fixed HeLa cells expressing FPs or tandemly-connected FPs. Amplitude-weighted fluorescence lifetimes were estimated from fluorescence lifetime decays by fitting with a bi-exponential. FRET efficiencies were then calculated from amplitude-weighted fluorescence lifetimes of the donor only and tandem-connected FPs". Are these FP pairs separated by the same length of the linker? Are these connected to CLC or not? Tagging FP pairs to the protein of interest changes the FRET efficiency due to rigidity, which influences dipole-dipole interactions between acceptor and donor.

12/There are many instances where the number of cells is stated but not the number of independent experiments (replicates). Have these experiments been performed only once? (Fig 2E, Fig 5D, Fig S2A, Fig S3 C and S3D, E and F, Fig S9C etc...)

The number of independent experiments should be at least $n=3$ which is not the case in Fig 3C, Fig 4C, Fig S12B, etc...

Minor comments:

- The vertical axis in several panels (i.e Figure 1D, 4C and Figure S4 etc...) are misleading and should start at 0.

-The sentence in line 762 (D) "Simulations on FRET efficiency between EGFP-CLC and near five CLC-ShadowY at different CLC-ShadowY expression levels" should be clarified. What does "near five" mean? How where the CLC-shadowY expression levels measured?

-typo on line 529, the word "device" is missing behind the word fracture

Reviewer #2:

Remarks to the Author:

Obashi et al. implement novel technology to study protein conformational changes at nanometer resolution in plasma membrane-localized cellular processes. Their methodology employs fluorescence lifetime imaging (FLIM) of FRET-paired fluorophores in tandem with platinum replica electron microscopy (PREM) to gain both small-scale spatial information from a diffraction limited spot, and larger-scale structural information from the same location within an unroofed human-derived cell. The authors apply this technique, dubbed "FRET-CLEM," to study conformational changes that occur in clathrin light chain (CLC) during the assembly of free clathrin triskelia into clathrin cages. The authors provide evidence that a conformational change distinct from changes previously observed in vitro occurs as the plasma membrane progressively bends from flat to spherical in shape, and that inhibition of this change prevents productive clathrin-mediated endocytosis (CME). We believe that this work not only provides novel insights into the mechanistic regulation of clathrin assembly, but it opens the door for acquiring long-awaited protein-level mechanistic information about the entire process in its native environment. In our opinion, this work merits publication in Nature Communications. We included suggestions below that we believe will strengthen this study.

Comments:

1. General comment about Results section headings: The Results section headings are in our opinion too vague and should summarize the findings of work described in each section. In particular, the first two section headings ("Correlative FLIM-FRET and PREM method (FRET-CLEM)" and "Conformational changes in CLC in cells") do not provide description of the main takeaways from Figures 1 and 2. An example of an improved section heading would be: "FRET-CLEM provides nanometer-scale spatial information at CCSs" for the first, and "The CLC N-terminus moves away from the clathrin triskelion vertex" for the second section, making it similar to the third heading, "CLC N-terminal region moves away from the plasma membrane."

2. Figure 2: While we like the idea of presenting proposed hypotheses with cartoons, the three models presented are not obviously connected to the final proposed model in Figure 6 (starting in an extended conformation and proceeding to the proposed new conformation). The final model in Figure 6 leads readers to believe the process starts with CLC in an extended conformation in vivo that changes to the new conformation as CME progresses, while the interpretation of data in Figure 1F leads readers to believe an extended conformation for CLC is not possible in vivo. The authors propose that because the full-length CLC data in 1D and the EED->QQN mutant data in 1F produce the same result, the CLC N-terminus isn't truly adopting an extended conformation in vivo as this depends on the N-terminal charged amino acids. This makes interpretation of Figure 6 confusing. It seems this is an important distinction between unassembled triskelia and flat clathrin lattices. It might help to add an "unassembled" column to Figure 2A before the "flat," "dome," and "sphere" columns and to continue this convention in Figure 6, and make note of this distinction in the text.

3. Supplemental epsin1-FRB construct results: We feel the inclusion of these data in the manuscript

needs to be better supported in the text. While the idea of an inducible membrane tether localized only to endocytic sites is compelling, overexpression of endocytic components can significantly alter the native process. For example, it's known that modulating epsin expression levels changes endocytic rate and productivity, which the authors mention in the text. Thus, it currently seems that the results from the overexpressed epsin1-FRB constructs at best provide the same interpretation as results from the PH-FRB constructs, and at worst make interpretation more confusing and confounded than the PH-FRB results because of epsin overexpression's effect on CME even before dimerization is induced.

4. Line 272: should read: "These results indicate that although overexpression of epsin1 had..."
Endogenous epsin1 should have no effect on endocytosis.

5. Line 366: affects -> affect (typo)

Some ideas for future experiments and controls for the paper:

1. The authors do a good job of explaining the limitations of their technique, one being potential CLC functional impairment by fluorescently tagging it. One way to potentially strengthen the conclusion from figure 3E that the CLC N-terminus moves away from the membrane more than the C-terminus would be to add a fluorescent protein to both ends of CLC. One of the fluorescent proteins would be EGFP, and the other would be a "dummy" protein of similar size and structure to EGFP that does not FRET with EGFP or DPA, and EGFP could be swapped to either the N or C terminus for FRET-CLEM. If this construct gives the same result, any concern that the differences seen between CLC-EGFP and EGFP-CLC are a result of how the protein was tagged rather than actual mechanistic differences, would be alleviated.

Response to Reviewers

We would like to thank the reviewers for their helpful and insightful comments on our manuscript. We appreciate the time and effort of these reviews. To address the reviewer's specific questions and comments, we have performed substantial new experiments and analysis and have revised the text, figures, and models accordingly. We believe that these additions and changes significantly strengthen the rigor of the manuscript. Below we provide a point-by-point response to all comments raised by the reviewers. We hope that the manuscript is now suitable for publication.

Reviewer #1 (Remarks to the Author):

In the manuscript entitled "A conformational switch in clathrin light chain regulates lattice structure and endocytosis at the plasma membrane of mammalian cells", Obashi and colleagues from the Taraska group develop a correlative fluorescence resonance energy transfer (FRET) and platinum replica electron microscopy method, named FRET-CLEM. They investigate the conformational changes in clathrin light chain by mapping distance changes both parallel and perpendicular to the plasma membrane. The authors find that the N-terminus of CLC moves away from the plasma membrane and triskelia vertex as lattices curve and that preventing this conformational switch increased clathrin structure size and inhibited endocytosis.

This work focuses on an important topic and overall, the experimental design/analysis is of high quality and the writing is good. From a methodological standpoint this work is original however the conclusions drawn are mitigated and not verified but other techniques leaving the possibility that the authors could be wrong. Also, in my opinion, the conclusions drawn are sometimes exaggerated and not controlled enough. For example, the schemes show that each one of the three light chains is tagged but in reality, the endogenous proteins occupy some of these positions and make the interpretations very different from the theoretical models drawn. Also, in the same line FRET results are difficult to interpret as expression levels of transfected constructs cannot be controlled. Overall, several problems detailed below will need to be addressed before publication in Nature communications:

Thank you for your positive evaluation and helpful comments on our manuscript. We hope the extensive changes and additions we have made to the manuscript have strengthened the conclusions.

Major comments (not listed in order of importance):

1/ One of the major claims in this work concerns the fact that the authors mention throughout the manuscript that there are three light chains for three heavy chains. Although this idea has been around in the field, it has been shown by the Blondeau group (Girard et al., 2005 MCP) using quantitative proteomics that this is not true in non-neuronal cells and that the clathrin heavy and light chains are expressed with a non-stoichiometric relationship. While this might seem trivial, in the particular work it is a major issue as it alters the interpretation of the FRET analysis.

Thank you for this important comment. We have cited the suggested paper and modified the abstract (line 12) and text (Line 36) accordingly.

To address this comment, we have made two important changes.

First, we clarify how the expression level of CLC probes, the presence of endogenous CLCs, and potential CLC-free heavy chains could influence the measured FRET efficiencies in a new cartoon scheme shown in supplementary Fig 4.

Second, we performed new analysis of FRET efficiencies between EGFP-CLC and CLC-ShadowY positions predicted from different in vitro models of assembled clathrin. In this analysis, the fluorescence lifetime of EGFP-CLC decreases through the FRET mechanism when CLC-ShadowY is located nearby. In the extended conformation of CLC predicted from the X-ray structures, three C-terminal probe positions (orange positions in the scheme shown in supplementary Fig 4a, left) are within distances of 45 Angstroms. In the case of the bent conformation predicted from the X-ray structures, five C-terminal positions (orange positions in the scheme shown in supplementary Fig 4a, right) could locate near EGFP and would induce detectable FRET decreases to lifetime if labeled with ShadowY. If we focus on a single heavy chain, there are thus four unique possible assembly states; 1) without CLC, 2) with endogenous CLCa or CLCb, 3) with EGFP-CLC binding, or 4) with CLC-ShadowY binding (supplementary Fig 4b). From these states, we calculated FRET efficiencies (supplementary Fig 4c). We only considered FRET among EGFP-CLC and the closest five C-terminal positions because other C-terminal positions would be outside the range of detectable FRET (greater than 150 angstroms). We also assumed that FP-attached CLCs are randomly incorporated throughout the clathrin lattices. This is a justified assumption from past work using super-resolution CLEM imaging (Sochacki et al. 2017). In this analysis we also include the probability of CHC binding to CLC-ShadowY as a CLC-ShadowY occupancy rate. This number in principle should mirror the expression levels of CLC-ShadowY and the FRET efficiency would increase as CLC-ShadowY occupancy increases. From these considerations, we find that the FRET efficiency of the extended conformation of the light chain is always larger than the bent conformation at all possible CLC-ShadowY occupancies. Thus, the conformational changes in CLC can be measured among CCSs across a range of expression levels (occupancies and stoichiometries) because all possible states show a decrease in FRET from the extended state to the bent state. We now describe these considerations in detail in figure legend (line 920-951).

Along with this analysis, we have also included new biochemical and imaging experiments (discussed below) to address the issue of expression level and labeling directly on our FRET measurements. We hope that these new explanations, modeling, experiments, and presentations improve the rigor and carefulness of our study.

2/ On line 182 the authors state “However, in these mutants, we found similar fluorescence lifetime changes as in the wild type protein (Fig 2D and f). These data do not support model 2”. The authors should also take into consideration the presence of endogenous CLC contribution in these clathrin-coated structures.

As discussed above, endogenous light chain is an important consideration in our experiments. To test for effects of endogenous light chain, we measured the degree of fluorescence lifetime changes among cells with a range of expression levels with additional FRET-CLEM experiments (supplementary Fig 11). Specifically, we quantitated cell average EGFP intensities in unroofed HeLa cells expressing EGFP-CLC, EGFP-CLCΔN, or EGFP-QQN, along with CLC-ShadowY (supplementary Fig 11a). All three conditions showed similar ranges in EGFP intensities and expression levels (supplementary Fig 11b). Next, we performed new FRET-CLEM experiments on these cells and obtained 13 new single-cell data sets for

each conditions. The EGFP intensity distributions spanned similar ranges (supplementary Fig 11c). Next, fluorescence lifetimes from single CCSs were grouped according to lattice structures (flat, domed and sphere) and were compared to the average values of flat structures (supplementary Fig 11d). For EGFP-CLC and EGFP-QQN, fluorescence lifetimes increased as clathrin lattices curved, regardless of the measured EGFP expression levels. The degree of fluorescence lifetime changes were small in EGFP-CLCΔN, matching our previous data. In summary, there was no measurable relationship between EGFP intensities (expression levels) and lifetime differences between spheres and flat structures for all three conditions (supplementary Fig 11e). Furthermore, the degree of differences were similar between EGFP-CLC and EGFP-QQN. Thus, we conclude that transfection levels were similar among different conditions and any differences in the expression levels as compared to endogenous protein levels did not affect the measured changes in FRET between clathrin curvature classes. These new control experiments and analysis support our previous arguments and models. We have updated the figures and text (line 198-199) accordingly. We hope these new experimental data improve the manuscript.

3/ Lifetimes depend on the local environment and rigidity.

We thank the reviewer for raising this point. Local environment can affect fluorescence lifetimes. For example, refractive index changes can influence fluorescence lifetimes (Suhling et al., 2002). We cited this paper and mentioned this potential complications in the discussion (line 403-404). This hyper-local effect is currently very difficult to test directly.

As the reviewer notes, probe flexibility is another important variable in the FRET equations. To directly test for linker flexibility of our probes, we now performed new anisotropy measurements using polarization-TIRF (pol-TIRF) microscopy with established orientation sensors (supplementary Fig 6). Specifically, the orientation of nucleoporins (Nups) in the nuclear pore complex were measured using pol-TIRF microscopy (Pulupa et al., 2020). Here, the short N-terminal α -helix of EGFP is conjugated to a C-terminal α -helix in the protein of interest by a continuous α -helical linker. When the number of amino acids in this helix is varied by one, two, or three residues, the chromophore rotates around the linker helix axis by angles dictated by α -helical geometry. Because the excitation dipole is relatively fixed within GFP, the p:s ratio [ratio in fluorescence intensity excited by pol-TIRF fields that are perpendicular (p-pol) or parallel (s-pol) to the coverslip] changes. However, if EGFP is conjugated by a flexible linker, the p:s ratio does not change according to linker length. To experimentally test this system, we first validated that we can reproduce the previously published results in our system. We imaged the bottom of the nucleus of HeLa cells expressing Nup54-EGFP494 constructs (Pulupa et al., 2020) (supplementary Fig 6a). Consistent with the previous work, the p:s ratio shifted by changing the linker length for the rigid linker probes but did not shift for the flexible linker probes (supplementary Fig 6b). Next, we applied this strategy to investigate the linker flexibility in our CLC probes. We conjugated EGFP to the C-terminal α -helix domain of CLC (at residue 206) by either rigid or flexible linkers (supplementary Fig 6c). We then imaged unroofed HeLa cells expressing CLC orientation sensors, CLC-EGFP, or EGFP-CLC (supplementary Fig 6d). The p:s ratio shifted with different lengths of the rigid linkers, but the p:s ratio was not changed by varying the lengths of the flexible linkers. The p:s ratio of CLC-EGFP and EGFP-CLC (which were used in our FRET experiments) were similar to those of the flexible linker sensors. These results support the idea that the linkers of CLC-EGFP and EGFP-CLC used in our FRET experiments are flexible and relatively orientation-insensitive. Furthermore, there was no significant correlation among EGFP intensity and the p:s ratio in CLC-EGFP and EGFP-CLC (supplementary Fig 6, e-g). These new experiments show that the

linkers in our clathrin probes are flexible across clathrin of all shapes and sizes. However, it is difficult to completely rule out some changes in linker flexibility. We now discuss this possible complication in text (line 391-394).

Do expression levels of these constructs play a role in clathrin lattice formation?

We have analyzed clathrin structures in transfected cells and non-transfected cells using platinum replica CLEM directly (supplementary Fig 18, originally supplementary Fig 14). Expression of the CLC probes did not change the size and number of clathrin lattice structures. This matches our past data where no changes were seen with expression of clathrin and many other proteins (Sochacki et al. 2017).

The authors need to provide the quantification of all these constructs expression levels by using for example Western blot experiments.

Thank you for this suggestion. To quantitate the expression levels of the transfected FRET constructs, we now include western blot analysis of CLC (supplementary Fig 7). From these data, the amount of CLCs in the entire cell increases 10 times by transfection and the ratio between transfected and endogenous CLCs is 25:1 for single transfection and 100:1 for double transfection (line 688-706). Thus, the vast majority of the CLCs should be attached to an FPs. According to these experiments, analysis, and modeling, we have changed the range of the fraction of CHC binding to ShadowY-attached CLC used in the simulations (supplementary Fig 4c and supplementary Fig 17). We hope these new experiments and analysis improve the manuscript.

Does the EGFP tagged C- terminal CLC construct show the same lifetimes?

Yes, along with the response to comment #5, we measured cell-average fluorescence lifetimes in unroofed HeLa cells expressing EGFP-CLC, CLC-EGFP, or EGFP-CLC and CLC-EGFP. All showed similar lifetimes.

4/ On line 99 the authors state that “We chose monomeric EGFP and a dark yellow fluorescent protein, ShadowY, as an optimized FRET pair by comparing six potential fluorescent protein (FP) pairs (Fig S2A)”. I assume all these FRET Pairs (donor and acceptors) are separated by the same length linker?

Thank you for this comment. We modified the Methods to clarify this text (line 445-448).

“For tandemly-connected FPs in supplementary Fig. 2a, N- (residues 1-5) and C-terminal disordered regions (residues 228-238 for EGFP and mTurquoise2, and 226-236 for mNeonGreen) of FPs were deleted. And two FPs were connected with a short linker (GGSGGS).”

are all these FRET pairs tagged to CLC? In the figure legend, it mentioned tandemly connected, more details are needed.

Thank you for this question. The probes shown in supplementary Fig 2a were expressed in the cytosol as controls to identify the maximum FRET efficiencies among different possible FP pairs. We did this to find the most efficient pair for our study and we felt including these data would be useful for the community. The FP pairs other than EGFP-ShadowY were not further used to study clathrin light chain. We have adjusted the figure legend to clarify this point (line 890).

“These probes were expressed in the cytosol.”

-EGFP- ShadowY-CLC (Fig 1A) I did not find the information about this construct in the methods section, not sure how well GFP and Shadow Y are separated

Thank you for this comment. We have modified the methods section to clarify this (line 448-449).

“For EGFP-ShadowY-CLC in Fig 1, tandemly-connected EGFP-ShadowY was fused to CLCa with a 10 a.a. linker.”

5/ What happens to the lifetime if expressing both CLC-EGFP and EGFP-CLC? are we expect an intermediate lifetime? Because stoichiometry might change, and homoFRET?

Fluorescence lifetimes of EGFP-CLC and CLC-EGFP were not different. Additionally, fluorescence lifetimes in HeLa cells expressing both EGFP-CLC and CLC-EGFP were similar to those of single-transfected cells (Revise Fig 1 in Comment #3). This is consistent with proposals where Homo-FRET does not affect fluorescence lifetime (Lakowicz, 2006).

6/ In the last part of the results section, the authors investigated whether manipulation of the CLC N-terminal position effects transferrin endocytosis in whole cells describe using Transferrin uptake as a functional assay which allows to quantification the clathrin-mediated endocytosis. While this is an important experiment, the amount of transferrin receptors at the surface could be altered in cells transfected with the different constructs and not reflect the true endocytic activity. One should know the expression levels of transferrin receptors upon expressions of different tagged CLC The authors should control this by showing that the transferrin receptor levels are unchanged and that uptake truly reflects endocytosis activity.

We thank the reviewer for this suggestion. To test whether the amount of transferrin receptor at the plasma membrane is changed by transfection, we performed new immuno-cytochemistry experiments. HeLa cells expressing FKBP and FRB probes were stained with anti-transferrin receptor antibody without permeabilization (supplementary Fig 16c). Staining intensities of surface transferrin receptors normalized to non-transfected cells in the same sample were compared (supplementary Fig 16d). The

transfection of FKBP and FRB probes did not change the amount of surface transferrin receptors. This is consistent with our previous data that transferrin uptake did not change with transfections (supplementary Fig 16b).

7/ Graphical representations of EGFP CLC, ShadowY CLC, and EGFP-ShadowY CLC are a bit confusing to the reader in Fig1 and Fig.2. Besides graphical representations, 2c is misleading because one should show all scenarios and combinations of donor and acceptor CLC, inter FRET, HomoFRET, and stoichiometry.

To make the differences in intra- and inter-molecular FRET more clear, we have now changed how the cartoon FRET positions are shown in Fig 1a and supplementary Fig 4b. To specifically highlight the changes in distances for a N-terminus of a specific CLC and the C-termini of surrounding CLCs, we now use the more distinctive star markers instead of circles (Fig 2c). Additionally we updated the text to describe all possible configurations (without CLC, endogenous CLCa or CLCb binding, EGFP-CLC binding, or CLC-ShadowY binding) and their possible effects on FRET in supplementary Fig 4 (line 153-157).

“Although the expression levels of the transfected probes, the ratio between endogenous and transfected CLCs, and the fraction of heavy chains binding to CLC could affect FRET efficiency, FRET efficiencies of the extended conformation are always larger than that of bent conformation when the degree of these factors are stable (Supplementary Fig. 4). This condition is satisfied when comparisons are made within a single cell.”

8/On line 332, the authors state that “The roles of other proteins or phosphorylation are alternate mechanisms that could drive this change“. What about the cargo effect? how does it change the CLC N-term configuration? Interactions with other proteins might lead to restriction of the dipole-dipole moment of the donor and acceptor, which affects the FRET. The authors need to provide the experimental evidence that shows protein-protein interactions does not influence the N-term conformational rearrangements.

Thank you for this comment. Cargo loading could possibly displace the N-terminal position from the lattice. We mentioned this possibility in the discussion (Line 346-347). Cargo could also change the network of protein-protein interactions in clathrin coats and could thus modify the N-terminal conformation of CLC. We also discuss this possibility (line 344-346).

Here, we investigated conformational changes in clathrin light chain during lattice curvature. We feel that changes due to cargo loading, protein-protein interactions, allostery, or post-translational modification in the light chain are all interesting and possible causes of this structural rearrangement. At present, we cannot show what mechanisms are directly responsible for generating these atomic changes in clathrin light chain. Future work is needed to address these points at the atomic scale.

9/ The present study focuses exclusively on HeLa cells. Although widely used, HeLa cells are cancerous with many intrinsic abnormalities and are notoriously known for forming large flat lattices. What about other cell types where various sizes of CCPs and structures are present in various cell types. It is very

important to test this proposed model in other cell types where the clathrin behavior is different such as for instance BSC-1 cells which only form large pits or neuronal cells which form smaller ones.

We previously published that almost all mammalian cell types we have imaged with EM contain flat clathrin lattices (Sochacki et al., 2021). Thus, we believe that the results and models from HeLa cells should extend to other cell types. We agree with the reviewer, however, that it is important to directly test our specific models in another cell type. Thus, we now include new FRET-CLEM data from SK-MEL-2 cells which have been well characterized and used in other studies (Doyon et al., 2011; Li et al., 2018). SK-MEL-2 cells expressing EGFP-CLC and CLC-ShadowY were analyzed by FRET-CLEM (supplementary Fig 9, a-c). Similar to HeLa cells (Fig. 2d), fluorescence lifetimes increased as clathrin lattices curved. Next, we performed FRET-CLEM on SK-MEL-2 cells expressing EGFP-CLC with DPA (supplementary Fig 9, d and e). Again, fluorescence lifetimes increased as clathrin lattices curved and the degree of changes was similar to that of HeLa cells (Fig. 3e). We concluded that our model of CLC conformational changes can be applied to SK-MEL-2 cells and likely other cultured mammalian cells that are commonly used. We hope these new experiments on another cell line help support our conclusions.

10/As mentioned in the discussion progress from flat clathrin lattice to spherical clathrin structures leads to differences in the orientations between donor and acceptor, which majorly affects the lifetime of the donor, and results in alterations in FRET efficiency.

As mentioned above, we performed new pol-TIRF measurements to examine FP flexibility and showed that our probes are flexible and insensitive to orientation. However, we cannot directly show that linker flexibility is not changed in all individual clathrin structures. We mention this possibility in the discussion (line 391-394). Furthermore, the fact that both FP-FP FRET and FP-DPA FRET both showed the same relative directions of movement across different structures and systems strongly supports the idea that the conformational changes we detect in the light chain are a result of distance changes in the protein and not confounding biophysical changes in or between the probes. We hope these data strengthen our conclusions.

Considering this without knowing the stoichiometry of CLC in clathrin-coated structures, lifetime based FRET efficiency leads to misinterpretations...

From the results of western blot (supplementary Fig 7 and comment #3), in our FRET experiments most of the CLCs should be labeled with a FP. We previously showed that the density of FP attached CLC in CCSs does not differ among clathrin structures with different curvatures (Sochacki et al., 2017). Thus, we feel that it is reasonable to propose that the degree of FP-tagged CLC binding to CHC in clathrin lattices is not different across clathrin structures. The results from HeLa cells expressing EGFP-CLC Δ N and CLC-ShadowY indicate that fluorescence lifetimes were similar across different lattice states (Fig. 2e) further supporting the idea that lifetime changes are not a result of changes in stoichiometry. Likewise, as mentioned above, the fact that both of our FRET systems (FP/FP and FP/DPA) showed the same type of FRET change supports our models. In the case of DPA-based FRET, the quantity of membrane-bound DPA should be constant across all possible states and in excess of the FP-based probes—thus insensitive to stoichiometry changes in the donor. We hope these new data strengthen the manuscript.

11/ On line 772, in the legend for FigS2: “FRET efficiencies of tandemly-connected fluorescent proteins

(FPs). Fluorescence lifetimes were measured in fixed HeLa cells expressing FPs or tandemly-connected FPs. Amplitude-weighted fluorescence lifetimes were estimated from fluorescence lifetime decays by fitting with a bi-exponential. FRET efficiencies were then calculated from amplitude-weighted fluorescence lifetimes of the donor only and tandem-connected FPs". Are these FP pairs separated by the same length of the linker? Are these connected to CLC or not?

As mentioned at comment #4, we have now modified the text to clarify these points. (line 445-452)

For tandemly-connected FPs in supplementary Fig 2a, N- (residues 1-5) and C-terminal disordered regions (residues 228-238 for EGFP and mTurquoise2, and 226-236 for mNeonGreen) of the FP were deleted. Here, the two FPs were connected with a short flexible linker (GGSGGS). As stated above, these probes were not further connected to CLC.

Tagging FP pairs to the protein of interest changes the FRET efficiency due to rigidity, which influences dipole-dipole interactions between acceptor and donor.

The FRET efficiency of EGFP-ShadowY (supplementary Fig 2a) and that of EGFP-ShadowY-CLC (supplementary Fig 3c) were similar. We believe that the GGSGGS linker between EGFP and ShadowY is flexible to allow for efficient conformational sampling and the pair of FP proteins was linked to the clathrin by 10 amino acids linker (LRSRAQASNS). However, throughout the manuscript, we avoid reporting absolute atomic distance due to the many potential variables raised by the reviewers including the issues of orientation.

12/There are many instances where the number of cells is stated but not the number of independent experiments (replicates). Have these experiments been performed only once? (Fig 2E, Fig 5D, Fig S2A, Fig S3 C and S3D, E and F, Fig S9C etc...)

The number of independent experiments should be at least n=3 which is not the case in Fig 3C, Fig 4C, Fig S12B, etc...

Thank you for this comment. We have now indicated the number of experiments (replicates) in the figure legends. Additionally, we have repeated experiments if less than 3 replicates were done previously. Now, all data presented are from at least 3 independent experiments. Our conclusions were not changed with additional data. We hope this is now acceptable.

Minor comments:

- The vertical axis in several panels (i.e Figure 1D, 4C and Figure S4 etc...) are misleading and should start at 0.

FLIM precisely measures small fluorescence lifetime changes. Thus, in most cases, the relative change is more important than the absolute lifetime values in relation to a lifetime value of zero. In this light, we scale our figures to clearly show lifetime changes in the ranges of the most salient data. Most published work present lifetime data in a similar fashion (Baumdick et al., 2018; Ma et al., 2017; Mercier et al., 2020; Sparks et al., 2018). When scaling to zero, these graphs can be very difficult to interpret in a figure for the reader.

-The sentence in line 762 (D) "Simulations on FRET efficiency between EGFP-CLC and near five CLC-ShadowY at different CLC-ShadowY expression levels" should be clarified. What does "near five" mean? How were the CLC-shadowY expression levels measured?

Thank you for this comment. In FRET simulations, we only considered FRET contributions among EGFP-CLC and the five closest C-terminal positions because the other C-terminal positions are too far to induce FRET (> 150 Angstroms). We moved this analysis to supplementary Fig 4c and modified the figure legends to help make our simulations more understandable.

-typo on line 529, the word "device" is missing behind the word fracture

Thank you. We have added this word.

Reviewer #2 (Remarks to the Author):

Obashi et al. implement novel technology to study protein conformational changes at nanometer resolution in plasma membrane-localized cellular processes. Their methodology employs fluorescence lifetime imaging (FLIM) of FRET-paired fluorophores in tandem with platinum replica electron microscopy (PREM) to gain both small-scale spatial information from a diffraction limited spot, and larger-scale structural information from the same location within an unroofed human-derived cell. The authors apply this technique, dubbed "FRET-CLEM," to study conformational changes that occur in clathrin light chain (CLC) during the assembly of free clathrin triskelions into clathrin cages. The authors provide evidence that a conformational change distinct from changes previously observed in vitro occurs as the plasma membrane progressively bends from flat to spherical in shape, and that inhibition of this change prevents productive clathrin-mediated endocytosis (CME). We believe that this work not only provides novel insights into the mechanistic regulation of clathrin assembly, but it opens the door for acquiring long-awaited protein-level mechanistic information about the entire process in its native environment. In our opinion, this work merits publication in Nature Communications. We included suggestions below that we believe will strengthen this study.

Thank you for your positive evaluation and helpful comments on our manuscript.

Comments:

1. General comment about Results section headings: The Results section headings are in our opinion too vague and should summarize the findings of work described in each section. In particular, the first two section headings ("Correlative FLIM-FRET and PREM method (FRET-CLEM)" and "Conformational changes in CLC in cells") do not provide description of the main takeaways from Figures 1 and 2. An example of an improved section heading would be: "FRET-CLEM provides nanometer-scale spatial information at CCSs" for the first, and "The CLC N-terminus moves away from the clathrin triskelion vertex" for the second section, making it similar to the third heading, "CLC N-terminal region moves away from the plasma membrane."

Thank you for this suggestion. We modified the headings in the results section.

“FRET-CLEM provides nanometer-scale spatial information at single clathrin sites” (line 99)

“CLC N-terminal region moves away from the clathrin triskelion vertex” (line 134)

2. Figure 2: While we like the idea of presenting proposed hypotheses with cartoons, the three models presented are not obviously connected to the final proposed model in Figure 6 (starting in an extended conformation and proceeding to the proposed new conformation). The final model in Figure 6 leads readers to believe the process starts with CLC in an extended conformation in vivo that changes to the new conformation as CME progresses, while the interpretation of data in Figure 1F leads readers to believe an extended conformation for CLC is not possible in vivo. The authors propose that because the full-length CLC data in 1D and the EED->QQN mutant data in 1F produce the same result, the CLC N-terminus isn't truly adopting an extended conformation in vivo as this depends on the N-terminal charged amino acids. This makes interpretation of Figure 6 confusing. It seems this is an important distinction between unassembled triskelia and flat clathrin lattices. It might help to add an “unassembled” column to Figure 2A before the “flat,” “dome,” and “sphere” columns and to continue this convention in Figure 6, and make note of this distinction in the text.

Thank you for this suggestion. We agree that it is important to clarify the differences in CLC conformations between unassembled triskelia in the cytosol or membrane and flat clathrin lattices in Fig 2. As suggested, we have now modified Fig 2a and the associated text (line 139-141).

“First, we assume that unassembled triskelion in the cytoplasm contains extended CLC and the spherical lattices in cells resembles in vitro assembled cages and contains bent CLC.”

3. Supplemental epsin1-FRB construct results: We feel the inclusion of these data in the manuscript needs to be better supported in the text. While the idea of an inducible membrane tether localized only to endocytic sites is compelling, overexpression of endocytic components can significantly alter the native process. For example, it's known that modulating epsin expression levels changes endocytic rate and productivity, which the authors mention in the text. Thus, it currently seems that the results from the overexpressed epsin1-FRB constructs at best provide the same interpretation as results from the PH-FRB constructs, and at worst make interpretation more confusing and confounded than the PH-FRB results because of epsin overexpression's effect on CME even before dimerization is induced.

Thank you for this comment. We agree that the interpretations of the epsin1-CLC tethering experiments are challenging due to the major functional role of epsin in endocytosis. To avoid this confusion, we have removed these data. In the future, it will be interesting to explore these functional effects of epsin (or many other proteins in the system) with this type of assay.

4. Line 272: should read: “These results indicate that although overexpression of epsin1 had...”
Endogenous epsin1 should have no effect on endocytosis.

We have removed the text related to the epsin1 experiments.

5. Line 366: affects -> affect (typo)

Thank you. We have fixed this error.

Some ideas for future experiments and controls for the paper:

1. The authors do a good job of explaining the limitations of their technique, one being potential CLC functional impairment by fluorescently tagging it. One way to potentially strengthen the conclusion from figure 3E that the CLC N-terminus moves away from the membrane more than the C-terminus would be to add a fluorescent protein to both ends of CLC. One of the fluorescent proteins would be EGFP, and the other would be a “dummy” protein of similar size and structure to EGFP that does not FRET with EGFP or DPA, and EGFP could be swapped to either the N or C terminus for FRET-CLEM. If this construct gives the same result, any concern that the differences seen between CLC-EGFP and EGFP-CLC are a result of how the protein was tagged rather than actual mechanistic differences, would be alleviated.

Thank you for this suggestion. Accordingly, we have made new dual EGFP and SNAP-tag attached single CLC probes (SNAP-CLC-EGFP and EGFP-CLC-SNAP). We then performed FRET-CLEM in HeLa cells expressing these probes with DPA (supplementary Fig 13). Similar to the original probes shown in Fig 3e, fluorescence lifetimes of EGFP-CLC-SNAP increased as lattices curved, and the degree of changes were larger than those for SNAP-CLC-EGFP. These results support our models of CLC conformational changes. Thus, we do not believe the fluorescent probes are impairing the conformational changes we detect. We hope these new data strengthen our findings.

We thank all the reviewers for their insightful and helpful comments. We hope the new experiments, analysis, and text have strengthened the manuscript.

References

- Baumdick, M., M. Gelleri, C. Uttamapinant, V. Beranek, J.W. Chin, and P.I.H. Bastiaens. 2018. A conformational sensor based on genetic code expansion reveals an autocatalytic component in EGFR activation. *Nat Commun.* 9:3847.
- Doyon, J.B., B. Zeitler, J. Cheng, A.T. Cheng, J.M. Cherone, Y. Santiago, A.H. Lee, T.D. Vo, Y. Doyon, J.C. Miller, D.E. Paschon, L. Zhang, E.J. Rebar, P.D. Gregory, F.D. Urnov, and D.G. Drubin. 2011. Rapid and efficient clathrin-mediated endocytosis revealed in genome-edited mammalian cells. *Nat Cell Biol.* 13:331-337.
- Lakowicz, J.R. 2006. Principles of Fluorescence Spectroscopy. Springer.
- Li, Y., M. Mund, P. Hoess, J. Deschamps, U. Matti, B. Nijmeijer, V.J. Sabinina, J. Ellenberg, I. Schoen, and J. Ries. 2018. Real-time 3D single-molecule localization using experimental point spread functions. *Nat Methods.* 15:367-369.
- Ma, Y., E. Pandzic, P.R. Nicovich, Y. Yamamoto, J. Kwiatek, S.V. Pigeon, A. Benda, J. Rosy, and K. Gaus. 2017. An intermolecular FRET sensor detects the dynamics of T cell receptor clustering. *Nat Commun.* 8:15100.
- Mercier, V., J. Larios, G. Molinard, A. Goujon, S. Matile, J. Gruenberg, and A. Roux. 2020. Endosomal membrane tension regulates ESCRT-III-dependent intra-luminal vesicle formation. *Nat Cell Biol.* 22:947-959.

- Pulupa, J., H. Prior, D.S. Johnson, and S.M. Simon. 2020. Conformation of the nuclear pore in living cells is modulated by transport state. *Elife*. 9.
- Sochacki, K.A., A.M. Dickey, M.P. Strub, and J.W. Taraska. 2017. Endocytic proteins are partitioned at the edge of the clathrin lattice in mammalian cells. *Nat Cell Biol*. 19:352-361.
- Sochacki, K.A., B.L. Heine, G.J. Haber, J.R. Jimah, B. Prasai, M.A. Alfonzo-Mendez, A.D. Roberts, A. Somasundaram, J.E. Hinshaw, and J.W. Taraska. 2021. The structure and spontaneous curvature of clathrin lattices at the plasma membrane. *Dev Cell*. 56:1131-1146 e1133.
- Sparks, H., H. Kondo, S. Hooper, I. Munro, G. Kennedy, C. Dunsby, P. French, and E. Sahai. 2018. Heterogeneity in tumor chromatin-doxorubicin binding revealed by in vivo fluorescence lifetime imaging confocal endomicroscopy. *Nat Commun*. 9:2662.
- Suhling, K., J. Siegel, D. Phillips, P.M.W. French, S. Lévêque-Fort, S.E.D. Webb, and D.M. Davis. 2002. Imaging the environment of green fluorescent protein. *Biophys J*. 83:3589-3595.

Reviewers' Comments:

Reviewer #1:

Remarks to the Author:

The authors have responded to most of my queries either with additional experiments or by answering to the comments. They have modified the text in the revised version to clarify the text and toned down some of their previous conclusions and now discuss some of the limitations. They have now added additional controls, performed additional transfection experiments and analysed more precisely the effect of CLC expression on the clathrin ultrastructure, have included western blot quantifications to control for their transfection experiments.

Overall, this is a much improved manuscript and I congratulate the authors for their rigorous work.

Reviewer #2:

Remarks to the Author:

The authors have done an excellent job addressing the reviews. We are fully satisfied with their responses.

Reviewer #3:

Remarks to the Author:

In this manuscript, Obashi et al have investigated with CLEM (FRET-FLIM and EM) conformational changes in clathrin light chain establishing the distances relative to the PM.

Overall all FRET-FLIM experiments were carried out with care and the authors know what they are doing. I have a couple of more suggestions to make, given the fact that it seems they have limited photons and are using the average lifetime approach (non-fitting).

As in all experiments using FPs one wonders if the insertion of these proteins have an impact on the overall structure and how the dipole-dipole interactions are affected by the relative orientation and not so much the distance. However, this are limitations that were discussed and in my opinion do not impact the conclusions of the paper.

--

Line 65: Other parameters that control FRET are dipole relative orientation and the overlap between the donor emission and acceptor absorptions spectra (on the top of dipole-dipole distances).

Figure 1. How many photons per pixel were employed to characterize the average lifetimes for the graphs? Which average lifetimes are you employing (the definition of the mean lifetime or the time or arrival per photon? If the latter, are you taking into account the background?

Figure 2.

Fig. 2d-f. What is measured here? FRET efficiency or apparent FRET efficiency? Why?

The FRET dynamic range is given in Figure1 in ns, to understand how much the lifetimes change as compared to this the lifetimes should be given together with either the true FRET efficiency or fD (or mfD).

One should measure the proportion of donor undergoing FRET (fD) or mfD with non-fitting approaches, to understand how many EGFP-CLC are engaged with CLC-ShadowY.

It would be worth to see these images and plot the lifetimes versus the number of photons to understand how the relative number of photons affect each measurement.

Figure 3.

Why the lifetimes are here now given again in ns? There should be some coherent way of showing how the lifetimes change as a function of FRET (always in ns, FRET efficiency (E) and fD).

Figure 4.

The FKBP12 system seems to work quite well, the dynamic change in lifetime diminution is good, but there seems to be a competition between two acceptors (DPA and miRFP). I guess my only question is why the authors have chosen to use miRFP (and infrared FP that will minimize the overlap between the donor emission and acceptor's absorbance) as compared to ShadowY for example. Also, here with two different acceptors the FRET efficiency is not possible to be quantified as one would have two different acceptors once AP21967 is added to the system.

Again not sure why the authors have chosen the increase or decrease in average lifetime instead of fD which indicates the proportion of donor bound to acceptor (mfD in the non-fitting approach).

--

Overall, it seems that the main conclusions of the paper come from FRET-FLIM experiments.

Supp Figure 3. The authors show how the FRET efficiency (true FRET efficiency?) changes as a function of photons, it seems that it increases when the photon budget is limited.

Please use these data to plot in a scattered plot how the average lifetime changes as a function of the photon count per pixel. There should not be a linear correlation. Provided that the authors claim they have not enough photons to fit, it would be of interest to see these graphs.

Supp Figure 5. It seems that overall you have a change of 2.2 ns for your EGFP-CLC alone to 2.1 ns when the acceptor is present (EGFP-CLC + CLC-ShadowY). 0.1 ns in average lifetime given the dispersion seems a little bit tight. However these differences are consistent. Provided that you can provide the graphs where you show that you were in a photon counting regime that did not affect the lifetime these results make sense to me. It is however worrying that cell 6 presents a higher lifetime (closer to the average lifetime of EGFP). Perhaps this cell was over-expressing or you had a better S/N there. Please explain.

--

The over-expression of EGFP-CLC will not solve the endogenous problem raised by reviewer 1. Lifetime is independent to the level of expression and one cannot extrapolate these results assuming that more photon counts mean less endogenous protein in the CCP/

The refraction index and the local environment does affect lifetime and it is possible that this effect is having an impact on the results. In fact, the lifetimes of the CLC-EGFP differ a lot from expressing eGFP alone in live cells ($\tau = 2.6$ ns). This alone indicates that there must be a crowding effect and big impact of the environment in all measurements. However, I do agree that all these conditions

being equal, the addition of an acceptor could allow to measure relative changes that in turn are related to FRET and CLC-CLC interactions.

Regarding stoichiometry and homoFRET heteroFRET cannot predict these questions. The authors should carry on other optical approaches to probe this (that would be indeed of interest to compare FFT NandB and stoichiometry with the FRET-FLIM results). Although this reviewer understands the complexity of these measurements.

Response to Reviewer #3

We would like to thank Reviewer #3 for their helpful and insightful comments on our manuscript. We appreciate the time and effort of the review and the thoughtful discussion on FLIM and FRET. We hope that our responses and changes have improved the manuscript. Below we provide a point-by-point response to all the comments raised.

Reviewer #3 report

In this manuscript, Obashi et al have investigated with CLEM (FRET-FLIM and EM) conformational changes in clathrin light chain establishing the distances relative to the PM. Overall all FRET-FLIM experiments were carried out with care and the authors know what they are doing. I have a couple of more suggestions to make, given the fact that it seems they have limited photons and are using the average lifetime approach (non-fitting). As in all experiments using FPs one wonders if the insertion of these proteins have an impact on the overall structure and how the dipole-dipole interactions are affected by the relative orientation and not so much the distance. However, these are limitations that were discussed and in my opinion do not impact the conclusions of the paper.

Thank you for your positive evaluation and helpful comments on our manuscript. We hope the changes and additions we have made to the manuscript have strengthened the conclusions and improved the rigor and carefulness of the paper.

Line 65: Other parameters that control FRET are dipole relative orientation and the overlap between the donor emission and acceptor absorption spectra (on the top of dipole-dipole distances).

Thank you for this comment. We have modified the text (line 63-65).

Figure 1.

How many photons per pixel were employed to characterize the average lifetimes for the graphs? Which average lifetimes are you employing (the definition of the mean lifetime or the time or arrival per photon? If the latter, are you taking into account the background?)

Thank you for this question. In Figure 1, we measured mean fluorescence lifetimes of single clathrin-coated structures identified from EM. We didn't measure the mean lifetime from each pixel in these structures. Photon counts per single clathrin-coated ROI in Figure 1 were $12,120 \pm 530$ (min-max: 4,324-19,068) for EGFP-CLC and $12,215 \pm 817$ (min-max: 2,629-44,616) for EGFP-ShadowY-CLC. The distributions of photon counts per single clathrin-coated structures are shown in Supplementary Figure 3g. Considering the average area of the measurement ROIs, average photon counts per pixel are 350-450/pixel. Average photon counts per pixel in FRET-CLEM experiments in Fig 2 are now presented in Supplementary Figure 8a.

Mean fluorescence lifetime is the average photon arrival time (line 533-535). Background was not subtracted. In our measurements, background counts were generally very low and signal-to-background ratio was >100 in most experiments (Supplementary Figure 8b). Mean fluorescence lifetime was not dependent on the signal-to-background ratio. We hope the addition of these data and clarification (line 536-539) aids in the interpretation and reproducibility of our data and analysis for the community.

Figure 2.

Fig. 2d-f. What is measured here? FRET efficiency or apparent FRET efficiency? Why?

We measured the differences in mean fluorescence lifetime among different curvature classes of clathrin-coated structures identified from the correlated EM images. As shown in Supplementary Figure 5, cell-average mean fluorescence lifetime varied among cells. This can be due to the differences in expression ratio of donor and acceptor probes and other factors. Therefore, to directly compare the different curvature class, we compared mean fluorescence lifetimes within individual cells. We discuss this detail in line 574-579.

The FRET dynamic range is given in Figure1 in ns, to understand how much the lifetimes change as compared to this the lifetimes should be given together with either the true FRET efficiency or fD (or mfD).

Thank you for this comment. To analyze as many clathrin-coated structures that span all the size and curvature states present in a cell, we used mean fluorescence lifetime instead of curve fitting. Reliable curve fitting requires more than 10,000 photons for a well-fit bi-exponential model and many of our structures had lower photon counts than this number (150 frames in Supplementary Figure 3g). This would have limited our analysis to only the brightest clathrin structures.

FRET in Fig1 (control intra-molecular FRET) and Fig2 (experimental inter-molecular FRET where there are possibly multiple acceptors, different distances, and different binding fractions across cells) arise from different conditions. Thus, we believe that a direct comparison between the absolute lifetime values shown in Fig1 and Fig2 is inappropriate.

One should measure the proportion of donor undergoing FRET (fD) or mfD with non-fitting approaches, to understand how many EGFP-CLC are engaged with CLC-ShadowY.

Thank you for this suggestion. As mentioned in the former responses, we compared the fluorescence lifetimes without fitting to investigate as many clathrin-coated structures with various sizes and curvatures as possible in an unbiased way. Without fitting, measured FRET efficiency and fD cannot be estimated adequately. In our experimental schemes, distances between donor and acceptor changes and the assumption of a two-component system (free and bound components) is not suitable. We, however, fully support the idea that additional quantitative analysis of FRET data is key for continued studies. In the future, it will be useful to find new methods to possibly increase the photon counts by using organic dyes, control the stoichiometry with novel labeling techniques, and apply new less noise-sensitive quantitative analysis. We have now added these points to the text (line 377-379, 386-388). We hope this is acceptable.

It would be worth to see these images and plot the lifetimes versus the number of photons to understand how the relative number of photons affect each measurement.

Thank you for this suggestion. We now plot the relationship between mean fluorescence lifetime and photon counts of single CCSs in a new figure (Supplementary Fig 8 c-h). There is no relationship between mean fluorescence lifetime and photon counts. We have added a new Supplementary Figure 8 and modified text to include these new analysis (line 172-174). We hope these new control data improve the manuscript.

Figure 3.

Why the lifetimes are here now given again in ns? There should be some coherent way of showing how the lifetimes change as a function of FRET (always in ns, FRET efficiency (E) and τ_D).

Thank you for this question. When we compared the lifetime between cells, we used “ns” (Fig 1d, 3c, and 4c). However, when we compared fluorescence lifetimes of clathrin-coated structures within single cells in FRET-CLEM, we used the difference in fluorescence lifetime (Fig 2d-f and 3e). We presented “ns” data in supplementary figures for all FRET-CLEM experiments. All the data were quantified with the same base units.

Figure 4.

The FKBP12 system seems to work quite well, the dynamic change in lifetime diminution is good, but there seems to be a competition between two acceptors (DPA and miRFP). I guess my only question is why the authors have chosen to use miRFP (and infrared FP that will minimize the overlap between the donor emission and acceptor’s absorbance) as compared to ShadowY for example.

As you mentioned, we used miRFP because the miRFP emission spectra and DPA absorption spectra do not overlap (Supplementary Fig 2c).

Also, here with two different acceptors the FRET efficiency is not possible to be quantified as one would have two different acceptors once AP21967 is added to the system.

Because we can neglect FRET between miRFP and DPA, we only consider FRET between EGFP and DPA and EGFP and miRFP. The distance between EGFP and miRFP will not change between the addition of DPA. Thus, the small amount of fluorescence lifetime change to EGFP induced by miRFP should be constant throughout the experiment. We track the relative direction of movement of EGFP to the plane of plasma membrane by measuring the change to EGFP fluorescence lifetime before and after adding DPA to the plasma membrane (Fig 4d). We agree with the reviewer that the quantification of FRET into distance is challenging in situations with many possible acceptors.

Again not sure why the authors have chosen the increase or decrease in average lifetime instead of τ_D which indicates the proportion of donor bound to acceptor ($m\tau_D$ in the non-fitting approach).

Thank you for this question. The FRET efficiency (fluorescence lifetime) changes as the distances between EGFP and DPA change. A two-component system (free and bound donor) cannot be assumed in EGFP-DPA FRET because the relative ratio between DPA and GFP is constant. Furthermore, the relative distance between EGFP and DPA changes. Thus, we present fluorescence lifetime data in the same format as the other figures and past work using DPA FRET.

Overall, it seems that the main conclusions of the paper come from FRET-FLIM experiments.

Supp Figure 3.

The authors show how the FRET efficiency (true FRET efficiency?) changes as a function of photons, it seems that it increases when the photon budget is limited.

Please use these data to plot in a scattered plot how the average lifetime changes as a function of the photon count per pixel. There should not be a linear correlation. Provided that the authors claim they have not enough photons to fit, it would be of interest to see these graphs.

Thank you for this comment. We have estimated FRET efficiency with fitting (line 515-529).

In Supplementary Figure 3e-g, unroofed membranes of HeLa cells expressing EGFP-ShadowY-CLC were imaged repeatedly (50 frames, 4 times). In panel e and f, normalized intensity (e) and FRET efficiency (f) are compared among four successive 50 frames sessions. In panel g, accumulated photon counts from 50, 100, 150, and 200 frames were measured. Fluorescence intensity did not change (e) and FRET efficiency slightly decreased (f) as the session number increased. Thus, differences in FRET efficiency are not due to the differences in photon counts.

We now plot the relationship between mean fluorescence lifetime and photon counts of single CCSs for each 50 frames sessions (Revise Fig 2). As the frame number increases, fluorescence lifetime increased even if the photon counts are similar. This is consistent with the cell-average data in panels e and f.

Here, we conclude that relative differences in photostability between EGFP and ShadowY on our microscope are likely the main cause of FRET changes in these control experiments instead of measurement limitations induced by the number of photons. We now described this experimental caveat in line 507-514.

Supp Figure 5.

It seems that overall you have a change of 2.2 ns for your EGFP-CLC alone to 2.1 ns when the acceptor is present (EGFP-CLC + CLC-ShadowY). 0.1 ns in average lifetime given the dispersion seems a little bit tight. However these differences are consistent. Provided that you can provide the graphs where you show that you were in a photon counting regime that did not affect the lifetime these results make sense to me.

Thank you for this comment. As suggested, we investigated the relationship between mean fluorescence lifetime and average photon counts within a CCS ROI or signal-to-background ratio in two new figures

(Supplementary Fig 8 a, b). Here, there is no clear relationship between mean fluorescence lifetime and average photon counts or the signal-to-background ratio. These data support the idea that FRET between EGFP and ShadowY corresponds to differences in mean fluorescence lifetimes. We have added a Supplementary Figure 8 and modified the text accordingly (line 172-174).

It is however worrying that cell 6 presents a higher lifetime (closer to the average lifetime of EGFP). Perhaps this cell was over-expressing or you had a better S/N there. Please explain.

We did not find clear relationship between mean fluorescence lifetime and photon counts or signal-to-background ratio (Supplementary Fig 8 a, b).

The over-expression of EGFP-CLC will not solve the endogenous problem raised by reviewer 1. Lifetime is independent to the level of expression and one cannot extrapolate these results assuming that more photon counts mean less endogenous protein in the CCP/

Thank you for this comment. We agree that fluorescence lifetime is independent of the expression level, yet the expression level affects the absolute FRET values in a particular cell. However, direct fluorescence intensity should reflect the amount of EGFP-tagged probes in a cell. In our western blot analysis in Supplementary Figure 7, higher expression of FP-tagged CLC decreases the amount of endogenous CLCs (comparing control, single, and double transfection). Thus, we believe that it is reasonable to assume that brighter cells express more FP-tagged CLC and the ratio between endogenous and FP-tagged CLC in these cells changes at clathrin-coated sites.

Most CLCs in transfected cells were tagged with FP (Supplementary Fig 7). There was no measurable relationship between EGFP intensities (expression levels) and lifetime differences between spheres and flat structures for all three probe pairs (WT, deltaN, and QQN) (Supplementary Fig 12). These data support the idea that stoichiometry differences do not affect our conclusion.

The refraction index and the local environment does affect lifetime and it is possible that this effect is having an impact on the results. In fact, the lifetimes of the CLC-EGFP differ a lot from expressing eGFP alone in live cells ($\tau = 2.6$ ns). This alone indicates that there must be a crowding effect and big impact of the environment in all measurements. However, I do agree that all these conditions being equal, the addition of an acceptor could allow to measure relative changes that in turn are related to FRET and CLC-CLC interactions.

Thank you for this comment. In our system, mean fluorescence lifetimes (average photon arrival times, not a time constant for exponential decay) of free EGFP in the cytosol were 2.16 ± 0.01 ns for living cells (Supplementary Fig 11) and 2.20 ± 0.01 ns for fixed cells (Supplementary Fig 2a). Therefore, we believe that differences in fluorescence lifetimes between past reported values and our values are due to experimental differences in the specific measurement systems or cells. As mentioned, all the measurements in the manuscript were conducted with the same conditions and the relative differences in lifetimes do not change the scientific conclusion of the manuscript.

Regarding stoichiometry and homoFRET heteroFRET cannot predict these questions. The authors should carry on other optical approaches to probe this (that would be indeed of interest to compare FFT NandB and stoichiometry with the FRET-FLIM results). Although this reviewer understands the complexity of these measurements.

Thank you for this suggestion. We agree that it is important to clarify the relationship among Homo/Hetero FRET and stoichiometry. In the future, it will be interesting to explore these relationship with the combination of other optical techniques in our FRET-CLEM system. We added this in discussion (line 377-379).

We thank all the reviewers for their insightful and helpful comments. We hope the new experiments, analysis, and text have strengthened the manuscript.